# Effects of forcing differences and initial conditions on inter-model agreement in the VolMIP volc-pinatubo-full experiment

Davide Zanchettin[1], Claudia Timmreck[2], Myriam Khodri[3], Anja Schmidt[4,5*], Matthew Toohey[6], Manabu Abe[7], Slimane Bekki[8], Jason Cole[9], Shih-Wei Fang[2], Wuhu Feng[10,11], Gabriele Hegerl[12], Ben Johnson[13], Nicolas Lebas[3], Allegra N. LeGrande[14,15], Graham W. Mann[10,11], Lauren Marshall[5], Landon Rieger[6], Alan Robock[16], Sara Rubinetti[1], Kostas Tsigaridis[15,14], Helen Weierbach[17,18]

[1]University Ca' Foscari of Venice, Department of Environmental Sciences, Informatics and Statistics, Via Torino 155, 30172 Mestre, Italy
[2]Max-Planck-Institute for Meteorology, Bundesstr. 53, 20146 Hamburg, Germany
[3]Laboratoire d'Océanographie et du Climat: Expérimentations et Approches Numériques, Institut Pierre-Simon Laplace, Sorbonne Universités /IRD/CNRS/MNHN, Paris, France.
[4]Department of Geography, University of Cambridge,Cambridge, U.K.
[5]Department of Chemistry, University of Cambridge, Cambridge, U.K.
[6]Institute of Space and Atmospheric Studies, University of Saskatchewan, Canada
[7]Research Institute for Global Change, Japan Agency for Marine-Earth Science and Technology, 3173-25 Showa-machi, Kanazawa-ku, Yokohama 236-0001, Japan
[8]Laboratoire Atmosphères, Milieux, Observations Spatiales, Institut Pierre-Simon Laplace, Sorbonne Université/CNRS/UVSQ, Paris, France
[9]Environment and Climate Change Canada, Toronto, ON, Canada
[10] University of Leeds, Leeds, UK
[11]National Centre for Atmospheric Science (NCAS-Climate), University of Leeds, UK
[12]University of Edinburgh,Edinburgh, U.K.
[13]Met Office, Exeter, U.K.
[14]NASA Goddard Institute for Space Studies, New York, NY, USA
[15] Center for Climate Systems Research, Columbia University, New York, NY, USA
[16] Department of Environmental Sciences, Rutgers University, New Brunswick, NJ, USA
[17]Earth and Environmental Sciences Area, Lawrence Berkeley National Laboratory, Berkeley, CA, USA
[18]Lamont Doherty Earth Observatory, Columbia University, New York, NY, USA

*now at: German Aerospace Center (DLR), Institute of Atmospheric Physics (IPA), Oberpfaffenhofen, Germany, and Ludwig-Maximilians University Munich, Metrological Institute, Munich, Germany

*Correspondence to*: Davide Zanchettin (davidoff@unive.it)

**Abstract.** This paper provides initial results from a multi-model ensemble analysis based on the volc-pinatubo-full experiment performed within the Model Intercomparison Project on the climatic response to volcanic forcing (VolMIP) as part of the sixth phase of the Coupled Model Intercomparison Project (CMIP6). The volc-pinatubo-full experiment is based on ensemble of volcanic forcing-only climate simulations with the same volcanic aerosol dataset across the participating models (the 1991-1993 Pinatubo period from the CMIP6-GloSSAC dataset). The simulations are conducted within an idealized experimental design where initial states are sampled consistently across models from the CMIP6-piControl simulation providing unperturbed

pre-industrial background conditions. The multi-model ensemble includes output from an initial set of six participating Earth system models (CanESM5, GISS-E2.1-G, IPSL-CM6A-LR, MIROC-E2SL, MPI-ESM1.2-LR and UKESM1).

The results show overall good agreement between the different models on the global and hemispheric scale concerning the surface climate responses, thus demonstrating the overall effectiveness of VolMIP's experimental design. However, small yet significant inter-model discrepancies are found in radiative fluxes especially in the tropics, that preliminary analyses link with

minor differences in forcing implementation, model physics, notably aerosol-radiation interactions, the simulation and sampling of El Niño-Southern Oscillation (ENSO) and, possibly, the simulation of climate feedbacks operating in the tropics. We discuss the volc-pinatubo-full protocol and highlight the advantages of volcanic forcing experiments defined within a carefully designed protocol with respect to emerging modeling approaches based on large ensemble transient simulations. We identify how the VolMIP strategy could be improved in future phases of the initiative to ensure a cleaner sampling protocol

with greater focus on the evolving state of ENSO in the pre-eruption period.

**Plain text summary**

This paper provides metadata and first analyses of the volc-pinatubo-full experiment of CMIP6-VolMIP. Results from six Earth system models reveal significant differences in radiative flux anomalies that trace back to different implementations of volcanic forcing. Surface responses are in contrast overall consistent across models, reflecting the large spread due to internal

variability. A second phase of VolMIP shall consider both aspects toward improved protocol for volc-pinatubo-full.

**1 Introduction**

The Model Intercomparison Project on the climatic response to Volcanic forcing (VolMIP, Zanchettin et al., 2016) defined a coordinated set of idealized volcanic perturbation experiments to be carried out in alignment with the protocol of the 6[th] phase of the Coupled Model Intercomparison Project (CMIP6, Eyring et al., 2016). VolMIP aims to assess the diversity of simulated

climate responses to large-magnitude explosive volcanic eruptions with global-scale sulfate aerosol and to identify the causes and processes that cause inter-model differences.

Accordingly, VolMIP experiments are based on multi-model ensemble simulations with coupled climate models that are designed upon two pillars. First, the prescribed stratospheric volcanic aerosol optical properties (often referred to as "forcing") used in the simulations must be the same across all participating models. Consensus volcanic forcing data sets were thus

defined for each experiment and implemented in terms of zonal and monthly mean optical properties of stratospheric volcanic aerosol. Second, climate conditions defining the initial states of individual members of the ensemble must be selected in a consistent manner across the participating models, so that the diagnosed expected climate response is not biased by potential effects of a preferential phase of ongoing internal variability at the time of the eruption (see, e.g., Zanchettin et al., 2013; Swingedouw et al., 2015; Lehner et al., 2016; Khodri et al., 2017; Coupe and Robock, 2021). This is achieved by defining

desired states of climate variability modes to be sampled along the parent CMIP6-DECK piControl simulation representative of unperturbed preindustrial climate conditions.

VolMIP Tier 1 experiments are branched in two main sets, named "volc-pinatubo" and "volc-long", respectively (Zanchettin et al., 2016). The volc-pinatubo experiments include a main experiment with full forcing (volc-pinatubo-full) and two sensitivity experiments, named volc-pinatubo-surf and volc-pinatubo-strat, aimed at disentangling the competing effects of the

two mechanisms known to determine the seasonal-to-interannual response to volcanic eruptions, i.e., surface cooling and stratospheric warming (see section 2.1.2 of Zanchettin et al., 2016, for details on volc-pinatubo-surf/strat). The experiments tackle uncertainty and inter-model differences in the climatic response to an idealized 1991 Mt. Pinatubo-like eruption, which is chosen as representative of the largest magnitude of volcanic events that occurred during the instrumental period to date. The volc-pinatubo experiments use a volcanic forcing dataset derived from satellite observations (Thomason et al., 2018) but

do neither account for the actual climate conditions at the time of the 1991 Mt. Pinatubo eruption nor for other forcing factors concomitant with the eruption, hence their idealized character. The idealized volc-pinatubo experiments complement full-forcing transient experiments, where historical climate simulations show a general improvement of CMIP6-generation Earth system models in the simulation of the high-latitude climate response to the 1991 Mt. Pinatubo eruption compared to previous CMIP results (Pauling et al., 2021).

This paper provides the first multi-model analysis of the volc-pinatubo-full ensemble. The aim is to provide a first assessment of the experiment and general guidance about emergent gaps of knowledge revealed by major discrepancies across results from different models. An additional aim is to assess the appropriateness of the VolMIP protocol and to propose improvements for a possible second phase of the initiative. Therefore, the paper first includes a short description of the experimental setup (Sect. 2.1) and an overview of the participating models (six at the moment of writing) including technical details of the

simulations such as the branching years from the CMIP6-DECK piControl (Sect. 2.2). Statistical techniques used in the multi-model analysis are described in Sect. 3, and results are presented in Sect. 4. Focus is on spatially integrated quantities regarding the energy fluxes at the top of atmosphere and at the surface, basic quantities describing global and hemispheric-scale surface climate parameters (temperature and precipitation). We illustrate how inter-model differences are largely reduced compared to previous analyses thanks to the application of the VolMIP protocol, although inconsistencies remain especially in post-

eruption radiative flux anomalies. First insights into dynamical responses are also proposed, where we illustrate how selected modes of large-scale climate variability and climate feedbacks show larger differences across models compared to surface climate responses. The impacts from sampling and ensemble size on post-eruption anomalies and the expected climate response are also illustrated. Sect. 5 discusses the main inconsistencies across models, provides guidance for future studies and suggests revisions to the volc-pinatubo-full experiment in a possible second phase of VolMIP.

## 2 Characteristics of volc-pinatubo-full

### 2.1 The experiment

The VolMIP protocol established the CMIP6 stratospheric aerosol dataset to be used for the volc-pinatubo-full experiment. This dataset covers the CMIP6 historical period (1850-2018) and provides zonal and monthly mean stratospheric aerosol extinction, single scattering albedo and asymmetry factor as a function of latitude, height, wavelength and time (Luo, 2018a, b). The recommended CMIP6 stratospheric aerosol forcing is tailored to the spectral resolution of each participating model, which can vary greatly. For the models included in this study, the number of solar bands ranges from 4 to 23, and the number of terrestrial bands ranges from 9 to 16. For the years around the 1991 eruption of Mt. Pinatubo, the CMIP6 stratospheric aerosol data is constructed directly from the Global Space-based Stratospheric Aerosol Climatology (GloSSAC) observational reconstruction (Thomason et al., 2018). GloSSAC is constructed mainly from satellite observations of stratospheric aerosol extinction. For the Pinatubo period, GloSSAC aerosol properties are retrieved from the SAGE II satellite instrument, with some gaps filled via tropical and mid-latitude ground- based lidar measurements, most notably due to the strong extinction by Pinatubo aerosols in the lower tropical stratosphere in the first months after the eruption. Gaps in high latitudes have been filled by regridding the observations onto equivalent latitude instead of geographical latitude. The CMIP6 volcanic aerosol forcing initially made available for CMIP6 historical simulations and the VolMIP volc-pinatubo-full experiment was labeled as version 3, and was based on GloSSAC version 1.0 (Thomason et al., 2018). Subsequently (in August 2018), an updated version (v4) was released, based on GloSSAC v1.1, which corrected erroneous aerosol extinction values which arose through cloud screening of satellite data in the lower stratosphere. Nonetheless, version 3 remains the recommended forcing data set for the VolMIP volc-pinatubo experiments and was used in all simulations included in this study. This choice was deemed to be preferable since inter-model consistency in forcing is one of the pillars of VolMIP and volc-pinatubo-full simulations were already performed by some modeling groups at the time version 4 was released. Therefore, in the case that CMIP6 historical simulations are run with the version 4 data set, inconsistency in the forcing contributes to explaining possible inconsistencies between the volc-pinatubo-full and the historical simulations performed with the same model. Comparison of stratospheric aerosol optical depth (SAOD) for the CMIP6 v3 and v4 data sets confirms that both data sets show a very similar spatiotemporal evolution of the SAOD, with an initial tropical peak, followed by transport of the aerosol to the mid and high latitudes of both hemispheres (Fig. S1). Differences between the two versions are strongest in the tropics in the first few months after the Mt. Pinatubo eruption, and at the highest latitudes of both hemispheres. Rieger et al., (2020) compared CMIP6-historical simulations performed with two models using v3 and v4 of the CMIP6 forcing dataset, and found generally small differences in the simulated post-Pinatubo surface climate response, with the notable exception of temperature response in the tropical stratosphere.

By protocol, a minimum of 25 simulations are branched-off from the parent piControl simulation on June 1$^{st}$ of selected years, sampled in a way that they include equally distributed cold/neutral/warm states of the El Niño-Southern Oscillation (ENSO) and negative/neutral/positive states of the North Atlantic Oscillation (NAO). By protocol, ranges are the lower tercile for the

negative/cold state, the mid-tercile for the neutral state, and the upper tercile for the positive/warm state. Beyond these indications, the protocol did not provide a specific recipe for the sampling algorithm. The recommended indices were the Niño3.4 sea-surface temperature (SST) index for ENSO, and the two-box index as used by Stephenson et al. (2006) for the NAO, both referring to winter-average data (DJF, with January as reference for the year) for the first post-eruption winter (i.e., including January 1992). Specifically, the NAO index was defined as the difference between spatial averages of 500 hPa geopotential heights over (20–55 N; 90W–60 E) and (55–90 N; 90W–60 E). Sampling of an eastern phase of the Quasi-Biennial Oscillation (QBO), as observed after the 1991 Pinatubo eruption, is preferred for those models that explicitly simulate this mode of stratospheric variability. The choice to limit the number of sampled climate modes to two is due to restrictions related to the ensemble size. ENSO and NAO were chosen due their relevance in the scientific literature concerned with short-term volcanically forced climate variability when the VolMIP protocol was defined.

## 2.2 The multi-model ensemble

The volc-pinatubo-full multi-model ensemble includes simulations from six models: the 5th version of the Canadian Earth System Model (CanESM5, Sect. 2.2.1), the NASA Goddard Institute for Space Studies Earth System model (GISS-E2.1-G, Sect. 2.2.2), the CMIP6 version of the Institut Pierre-Simon Laplace coupled atmosphere-ocean general circulation model (IPSL-CM6A-LR, Sect. 2.2.3), the Model for Interdisciplinary Research on Climate, Earth System version2 for Long-term simulations (MIROC-ES2L, Sect. 2.2.4), the Max Planck Institute Earth System Model version 1.2 in its low resolution version (MPI-ESM1.2-LR, Sect. 2.2.5) and the 1st version of the UK Earth System Model (UKESM1, Sect. 2.2.6). All six models are in their exact CMIP6 version as employed for the CMIP6-endorsed initiative VolMIP.

The main characteristics of the participating models are reported in Table 1. Among the reported characteristics is each model's equilibrium climate sensitivity (ECS). ECS is relevant here due to a rich debate in the scientific literature about the use of climate anomalies after volcanic eruptions to infer equilibrium climate sensitivity (ECS) or transient climate sensitivity (e.g., Wigley et al., 2005; Boer et al., 2007; Merlis et al., 2014).

## 2.2.1 CanESM5

The fifth version of the Canadian Earth System Model (CanESM5) couples together models of the atmosphere (CanAM5), land (CLASS), ocean/sea-ice (CanNEMO), atmospheric carbon cycle (CTEM) and ocean biogeochemistry (CMOC) which are briefly described in Swart et al. (2019). The atmosphere is resolved using a T63 horizontal resolution, roughly 2.8° in longitude and latitude, and 49 vertical levels while the ocean is horizontally resolved at roughly 1° and 45 levels. The version of CanESM5 used in this study is CanESM5.0.3, described in Swart et al. (2019). A 1000-year long piControl simulation using CanESM5 is generally stable for heat, water and carbon related quantities, with observed drifts in some variables, e.g., ocean carbon flux, that are much smaller than anthropogenic signals. The ECS of CanESM5 is greater than CanESM2 (used for CMIP5) with a value of 5.67 K compared with 3.7 K (Zelinka et al., 2020) which is mainly attributed to changes in cloud feedbacks (Virgin et al., 2021).

### 2.2.2 GISS-E2.1-G

The NASA Goddard Institute for Space Studies Earth System modelE version 2.1, coupled to the GISS ocean (GISS-E2.1-G, Kelley et al., 2020), is one of several versions of the NASA GISS climate model submitted to CMIP6. It uses a regular grid across both the ocean and atmosphere. It has a 1.25° in longitude by 1° in latitude, 40-layer ocean coupled to a 2.5° in longitude by 2° in latitude 40-layer atmosphere with a top at 0.01 mb. The non-interactive version of atmospheric aerosols and chemistry is used in VolMIP (physics version 1). Volcanic eruptions are represented by specifying monthly averaged volcanic sulfate (optical properties described in Lacis et al., 1992) aerosol optical depth in 69 layers (5 km-39.5 km) and 36 latitude bands with no zonal variation (Thomason et al., 2018); the general implementation strategy of the GISS stratospheric volcanic aerosols are described in Sato et al. (1993), which uses a lower resolution version of the GISS model as well as a coarser resolution stratospheric aerosol boundary condition. ENSO variability is relatively large in this version of the model with variance about 50% greater than that observed (Kelley et al., 2020). ECS is about 3.6 °C for doubling of $CO_2$, with remarkable consistency to previous versions of the model. Further information about GISS-E2.1-G can be retrieved at the url: https://data.giss.nasa.gov/modelE/cmip6/

### 2.2.3 IPSL-CM6A-LR

The CMIP6 version of the Institut Pierre-Simon Laplace (IPSL) coupled atmosphere-ocean general circulation model is the low-resolution IPSL-CM6A-LR (Boucher et al., 2020) version 6.1.0, corresponding to a grid resolution of its atmospheric component LMDZ6A-LR of 1.25° in latitude and 2.5° in longitude and 79 vertical levels (Hourdin et al., 2020). LMDZ6A-LR is coupled to the ORCHIDEE (d'Orgeval et al., 2008; Cheruy et al., 2020) land surface component, version 2.0. In IPSL-CM6A-LR, the oceanic component uses the Nucleus for European Models of the Ocean (NEMO), version 3.6 (Madec et al., 2017), which includes other models to represent sea-ice interactions (NEMO-LIM3; Vancoppenolle et al., 2009; Rousset et al., 2015) and biogeochemistry processes (NEMO-PISCES; Aumont et al., 2015). Compared to the 5A-LR model version and other CMIP5-class models, IPSL-CM6A-LR was significantly improved in terms of the climatology, e.g., by reducing overall SST biases and improving the latitudinal position of subtropical jets. The IPSL-CM6A-LR is also more sensitive to $CO_2$ forcing increase (Boucher et al., 2020) and represents a more robust global temperature response than the previous CMIP5 version consistently with current state-of-the-art CMIP6 models (Zelinka et al., 2020).

### 2.2.4 MIROC-ES2L

The Model for Interdisciplinary Research on Climate, Earth System version2 for Long-term simulations (MIROC-ES2L) consists of the coupled atmosphere-ocean general circulation model called MIROC5.2, the land biogeochemistry component (VISIT), and the ocean biogeochemistry component (OECO2) (Hajima et al., 2020). The horizontal resolution of the atmosphere and the land is set to have T42 spectral truncation, which is approximately 2.8° intervals for latitude and longitude. The atmospheric vertical resolution is 40 layers up to 3 hPa. The horizontal grid of the ocean model is built on a tripolar system

that is divided horizontally into 360×256 grid points. (To the south of 63° N, the longitudinal grid spacing is 1° and the meridional spacing becomes fine near the Equator. In the central Arctic Ocean, the grid spacing is finer than 1° because of the tripolar system.) The ocean model has 62 vertical levels. VISIT simulates carbon and nitrogen dynamics on land. The OECO2 is a nutrient–phytoplankton–zooplankton–detritus-type model that is an extension of the previous model, MIROC-ESM (Watanabe et al., 2011). The effective climate sensitivity in MIROC-ES2L is lower than the previous version of MIROC-ESM (4.7 °C for MIROC-ESM, see Andrews et al., 2012, and 2.7 °C for MIROC-ES2L, see Tsutsui, 2020).

### 2.2.5 MPI-ESM1.2-LR

The Max-Planck-Institute Earth-System-Model (MPI-ESM) is composed of four components: the atmospheric general circulation model ECHAM6 (Stevens et al., 2013), the ocean-sea ice model MPIOM (Jungclaus et al., 2013), the land component JSBACH (Reick et al., 2013), which is directly coupled to ECHAM6, and the ocean biogeochemistry model HAMOCC (Ilyina et al., 2013), which is directly coupled to MPIOM. VolMIP experiments are performed with the MPI-ESM version 1.2 (version number mpiesm-1.2.01p6, see: Mauritsen et al., 2019) in its low resolution (MPI-ESM1.2-LR). In MPI-ESM1.2-LR, ECHAM6.3 is run with a horizontal resolution of T63 (~200 km) and 47 vertical levels, while MPIOM is run with a nominal horizontal resolution of 1.5° and 40 vertical levels. ECHAM6.3 includes modifications of the convective mass flux, convective detrainment and turbulent transfer, the fractional cloud cover and a new representation of radiative transfer with respect to its CMIP5 version (Stevens et al., 2013), while MPIOM remained largely unchanged with respect to the CMIP5 version of MPI-ESM (Jungclaus et al., 2013). A detailed description of all MPI-ESM1.2 updates is given in Mauritsen et al. (2019)., which contains ECHAM6.3. Climate sensitivity in the MPI-ESM1.2 was tuned to match the instrumental record warming by targeting an ECS of about 3 K using cloud feedbacks.

### 2.2.6 UKESM1

The first version of the UK Earth System Model (UKESM1) is fully described by Sellar et al. (2019a), the scientific and technical implementation within the CMIP6 experiments described in Sellar et al. (2019b). UKESM1 differs from its predecessor HadGEM2-ES (Collins et al., 2011) in being developed via a partnership between the UK Met Office and the UK Universities (funded via the UK Natural Environment Research Council). UKESM1 is built around the physical atmosphere-ocean climate model HadGEM3-GC3.1 (Kuhlbrodt et al., 2018; Williams et al., 2018), combining the Global Atmosphere 7.1 (GA7.1) configuration of the UK Met Office Unified Model (Walters et al., 2019; Mulcahy et al., 2018) with the NEMO ocean model (Storkey et al., 2018), the CICE sea-ice model (Ridley et al., 2018) and the JULES land-surface model (Best et al., 2011). Two major developments since HadGEM2-ES include the atmosphere physical model having a well-resolved stratosphere with stratosphere-troposphere chemistry (Archibald et al., 2020) and tropospheric aerosol radiative forcings from the GLOMAP-mode modal aerosol microphysics module (Mann et al., 2010; Bellouin et al., 2013). Another important progression since HadGEM2-ES is that the UKESM1 terrestrial biogeochemistry module within JULES (Clark et al., 2011) has coupled carbon and nitrogen cycles, also with enhanced land management. The main characteristics and behaviour of the

UKESM1 deck simulations for CMIP6 (pre-industrial control, abrupt 4xCO2, 1% increasing CO2 and post-industrial historical) are presented in Sellar et al. (2019a).

## 2.2.7 Forcing implementation

The VolMIP protocol recommends volcanic forcing input data below the model tropopause to be replaced by climatological or other values of tropospheric aerosol used by the models (see Zanchettin et al., 2016). This is a potential source of inter-model disagreement already at the forcing level, given differences across models in the vertical structure of the simulated atmosphere and choices made regarding the definition of the tropopause and the climatological reference value of tropospheric aerosol. The presence of forcing differences across participating models is illustrated in Fig. 1 by monthly values of the aerosol optical thickness at 550 nm due to stratospheric volcanic aerosols (variable aod550volso4[1]) averaged over the tropics (30°S-30°N). The amplitude of forcing differences between models vary through time, as can be seen for instance comparing results from IPSL-CM6A-LR and MPI-ESM1.2-LR, which match at the peak of the forcing but differ appreciably during the decaying phase. The largest difference occurs during the initial steep aerosol rise. Differences also concern the stratospheric background aerosol, which is present in MPI-ESM1.2-LR, GISS-E2.1-G and UKESM1 but not in other models (note, MPI-ESM1.2-LR volc-pinatubo-full simulations started in January 1991).

## 2.2.8 Initial conditions

Following the VolMIP protocol, branching from the piControl simulations was designed to sample combined states of NAO and ENSO in such a way that a broad range of internal unperturbed variability is considered at the time of the peak of the applied forcing, i.e., during the first post-eruption winter (DJF 1992, with January setting the reference for the year). The approach therefore aims at selecting climate conditions at the time of the eruption that are preconditioning a broad variety of states of ENSO and NAO in the following winter, to determine whether the climate response is influenced by developing anomalies in such modes.

Figure 2 illustrates the sampled winter average ENSO and NAO states from the piControl simulations for all models. For both modes, indices are standardized where the original DJF time series is modified by removing the long-term average calculated over the whole piControl and then by dividing it by the square root of the variance calculated over the whole piControl. For all models, the sampling complies with the general scope of VolMIP as the homogeneous spread of the scatterplots encompassing all quadrants confirms that different unperturbed coupled states of ENSO and NAO are considered corresponding to the first post-eruption winter (DJF 1992, Fig. 2a). Generally, the VolMIP sampling protocol regarding equally distributed sampling across first, second and third terciles of an index is satisfied, with the noticeable exception of a biased sampling of warm versus cold ENSO in MIROC-ES2L (see numbers in legend of Fig. 2). Still, some differences across models are apparent, for instance the range of sampled ENSO states in the standardized DJF Niño3.4 index is comparatively smaller

---

[1] Variable names are reported as defined by the CMIP6 "Climate Model Output Rewriter"

in CanESM5 than in other models and there is a small negative ENSO bias in MPI-ESM1.2-LR and UKESM1. The inter-model differences reflect the application of different sampling algorithms and/or subjective choices. Concerning the first point, initial states of MIROC-ES2L were sampled at 200-year intervals without an explicit consideration of the corresponding ENSO and NAO states. Then, the circular structure emerging for some models in the NAO-ENSO scatterplot corresponding to the last pre-eruption boreal winter (DJF 1991, Fig. 2b) reveals how some modeling groups (CanESM5, IPSL-CM5A-LR and MPI-ESM1.2-LR) targeted the last pre-eruption boreal winter and not the first post-eruption boreal winter, for the selection of initial states. This was done using a sampling algorithm yielding the visible circular structure. The sampling strategy is further discussed in Sect. 4.5.

The simulations are then started on May 31st of the year of the Pinatubo eruption for all models except MPI -ESM1.2-LR, for which the simulations are started on January 1st of the year of the eruption due to technical reasons. This study uses an ensemble of 25 simulations - the minimum ensemble size set by the protocol - for each contributing model, unless otherwise specified. Note that more realizations are available from certain models, for instance 121 realizations are available for GISS-E2.1-G and 40 realizations are available for CanESM5. The realizations considered here are those labeled from r1 to r25 in the metadata. Supplementary Table S1 provides additional information about the volc-pinatubo-full and piControl simulations used in this study.

## 3 Statistical methods and diagnostics

Climate responses to the volcanic forcing are quantified as anomalies with respect to the unperturbed climatology in each model. These are differences between the output of each realization in the volc-pinatubo-full multi-model ensemble and the climatology calculated for the whole length of the piControl, including seasonality. The anomaly method assumes that the unperturbed climate is characterized by uncorrelated white noise, which adds to the forced response in the volc-pinatubo-full simulations. Accordingly, the expected forced response can then be calculated as the ensemble mean anomaly whereas the spread of anomalies reflects noise. The method smears out climate variations on seasonal or longer timescales that emerge in the volc-pinatubo-full simulations and may have already been in progress in the piControl at the time chosen to start the volc-pinatubo-full simulation and might therefore be included in the calculation of the response. For instance, such variations can be due to ocean dynamics or sea-ice changes. This could be relevant, for instance, considering the biases in the averaged sampled states of ENSO in some models (Fig. 2a). Additional approaches to the quantification of the climate responses have been tested, including calculation of paired anomalies, i.e., deviations of the volc-pinatubo-full realizations from the corresponding branch of the piControl. These alternative methods occasionally yield different results compared to the anomaly method shown in the main results concerning the expected climate response, i.e., the ensemble mean, and the evaluation of statistical significance of inter-model differences. Accordingly, different approaches are discussed whenever deemed necessary.

The significance of inter-model differences in the multi-model ensemble is estimated based on the Mann-Whitney U test where the ensemble of each model is tested against the aggregated multi-model ensemble of the other models.

The analysis is performed on monthly values of selected relevant diagnostics spatially averaged over four regions. These are the full globe (hereafter GL), the Northern Hemisphere extratropics (30°-90°N, hereafter NH), the tropics (30°S-30°N, hereafter TR), and the Southern Hemisphere extratropics (30°-90°S, hereafter SH).

The VolMIP protocol refers to the Nino3.4 index as a diagnostic for the state of ENSO. However, the Nino3.4 index is known to be an unreliable diagnostic to detect post-eruption ENSO variability (e.g., Khodri et al., 2017). Therefore, ENSO responses are diagnosed also using the relative SST Nino3.4 (RSST-Nino3.4) index. The RSST-Nino3.4 index is calculated as the Nino3.4 index but using SST anomalies calculated as deviations from the tropical SST average (over the latitudinal band from 20°S to 20°N) instead of absolute SSTs.

A simple assessment of the global upward LW radiation flux across the atmospheric column and of the cloud-albedo feedback in the tropics and their dependence on the mean state of the unperturbed climate is performed (Sect. 4.4). Specifically, the atmospheric LW transmittance is diagnosed through the ratio LWt/LWs↑, where LWt is the global average top-of-atmosphere LW radiation (rlut) and LWs↑ is the global average upward LW radiation at the surface (rlus) (e.g., Zanchettin et al., 2013). Cloud-albedo interactions in the tropics are diagnosed through the ratio SWt/SWtcs, where SWt (rsut) and SWtcs (rsutcs) are the top-of-atmosphere upward solar radiation under full-sky and clear-sky conditions, respectively, over the TR region. For both diagnostics, values are calculated for the volc-pinatubo-full simulations and for the corresponding piControl sections, then the ratio between them is calculated, e.g., (LWt/LWs↑)$_{\text{volc-pinatubo-full}}$ / (LWt/LWs↑)$_{\text{piControl}}$. Values below one of the ratios are associated with a strengthening of the underlying processes and feedbacks. For instance, for atmospheric LW transmittance, a value below one of the abovementioned ratio implies (LWt/LWs↑)$_{\text{volc-pinatubo-full}}$ being less than (LWt/LWs↑)$_{\text{piControl}}$ , hence a decrease of atmospheric LW transmittance under volcanic forcing compared to unperturbed conditions. These diagnostics integrates the effects of diverse processes, including direct radiative responses forced by the volcanic aerosol and feedbacks operating through changes in the global-mean surface temperature. Hence, they do not allow for a separation between direct and indirect effects of volcanic forcing, which requires additional experiments such as volc-pinatubo-surf/strat.

The effect of ensemble size on the uncertainty estimation in the expected climate response (i.e., ensemble mean) is quantified for each model through changes in the standard error of the ensemble mean calculated for different ensemble sizes (Sect. 4.6). In practice, for each ensemble size from 3 to 25, all possible permutations of the full ensemble for the considered size are retrieved. Then, for each size and sub-ensemble obtained from the permutations, the standard error of the ensemble mean is calculated for the anomalies of the variable of interest, i.e., the square root of the variance of the sub-ensemble anomalies divided by the square root of the sub-ensemble size is calculated. Then, means and 5th and 95th percentiles of the so-obtained standard errors for each ensemble size are plotted. The standard error is calculated for anomalies of annual mean values of the year 1992 for two key surface variables: global-mean near-surface air temperature and global-mean precipitation.

The volc-pinatubo-full output for GL, TR, NH and SH near-surface air temperature and precipitation is compared with observational data. Gridded data and regional time series from the HadCRUT.5.0.1.0 analysis covering the period 1850-2021 are used for near-surface air temperature (Morice et al., 2021). For GL, the analysis time series including ensemble mean and uncertainty is used, for TR, NH and SH only the ensemble mean is calculated from the gridded ensemble mean data. The monthly gridded Global Precipitation Climatology Project (GPCP) Version 2.3 Combined Precipitation Data Set covering the period 1979-2021 is used for precipitation (Adler et al., 2003). The GPCP precipitation data are provided by the NOAA/OAR/ESRL PSL, Boulder, Colorado, USA, from their web site at https://psl.noaa.gov/data/gridded/data.gpcp.html (last accessed on 02/02/2022). GPCP data are pre-processed to remove the climatological seasonal cycle. Both HadCRUT and GPCP time series are detrended with a second-order polynomial fit to the data to remove long-term variability, especially the centennial global warming signal, and improve comparability with the volc-pinatubo-full output. Observed anomalies are defined as deviations from the 1990 average.

## 4 Results

### 4.1 Mean state and variability in piControl

The simulated mean state and variability of piControl, i.e., under unperturbed conditions, can affect the post-eruption response through excitation of internal climate modes by the eruption, which in turn also depends on their amplitude and phase at the time of the eruption (see Fig. 2 and Sect. 4.3), and/or through controlling the strength of climate feedbacks that operate through changes in the global-mean surface temperature (see Sect. 4.4). The climate state in piControl is illustrated in the form of Box-Whisker plots of relevant diagnostics calculated for the whole length of the simulations (see Supplementary Table 1), including global and regional averages of annual-mean near-surface air temperature (Fig. 3a-d), and winter ENSO and NAO indices as defined by the VolMIP protocol (Fig. 3e,f).

There are substantial differences in the simulated near-surface air temperature across models: IPSL-CM6A-LR has an overall cooler climate compared to the other models, of about 1°C in the global-mean surface temperature compared to CanESM5, GISS-E2.1-G and MPI-ESM1.2-LR; in contrast, UKESM1 and MIROC-ES2L have a warmer climate, of more than 1°C compared to other models. The colder global conditions in IPSL-CM6A-LR stem mostly from colder tropics and southern extra-tropics, for the latter, even colder conditions than IPSL-CM6A-LR are found for CanESM5. However, IPSL-CM6A-LR yields substantially warmer and less variable winter sea-surface temperature in the equatorial Central Pacific (Nino3.4 region) compared to other models, possibly reflecting different biases in the models. These differences may affect climate feedbacks as well as dynamical responses. The warmer climate of MIROC-ES2L stems from substantially warmer extra-tropical regions compared to other models, which may affect the response in terms of, among other processes, meridional energy transports and sea ice-albedo feedback. In contrast, the warmer climate of UKESM1 mostly stems from a warmer tropical region compared to other models, which may affect especially cloud-albedo feedbacks operating there.

The DJF Niño3.4 index, which is used to illustrate ENSO, shows substantial differences in the distributions across individual models. IPSL-CM6A-LR yields a warmer mean state of ENSO (above 28 °C) and a smaller variance of ENSO compared to other models. The distribution of ENSO in IPSL-CM6A-LR does not overlap with those of CanESM5, MPI-ESM1.2-LR and

UKESM1, whose mean state of ENSO is below 26 °C. ENSO shows a skewed distribution with a long tail toward strong El Niño events in MIROC-ES2L compared to the other models that yield rather symmetric distributions for the Niño3.4 index. There are similar differences across models concerning the NAO in terms of both mean state and variability of the mode. MIROC-ES2L displays a lower mean value of the non-standardized NAO index, indicating a smaller mean difference between boxes hence a smaller meridional gradient in the 500 hPa geopotential height, and a smaller variance compared to other models.

**4.2 TOA and surface radiative fluxes**

Left panels in Fig. 4 show the net top-of-atmosphere (TOA) vertical radiative fluxes calculated as anomalies of incoming shortwave (rsdt) minus outgoing shortwave (rsut) minus outgoing longwave radiation (rlut). At the global scale, the models agree on a largest average negative anomaly (i.e., reduced downward flux) of about 2 Wm-2 occurring in the first post-eruption boreal winter and on a persistence of the volcanic perturbation to TOA fluxes until 2.5 years after the eruption, i.e., until the

third post-eruption boreal winter. The ensemble spread is also largely consistent across models. There are inter-model differences during the first six post-eruption months, with models clustering into two groups, with slower increase of (hence smaller) radiative anomalies in IPSL-CM6A-LR, CanESM5, UKESM1 and MIROC-ES2L, and a faster increase of (hence larger) radiative anomalies in MPI-ESM1.2-LR and GISS-E2.1-G, with differences between clusters exceeding 0.5 Wm-2. The models agree remarkably on the extra-tropical response, whereas a significantly weaker response is found for IPSL-

CM6A-LR in the tropics from the time of the eruption to mid 1992. The smaller TOA net flux anomaly in IPSL-CM6A-LR is produced by a combination of the LW and SW components, especially in the tropics (see supplementary Figs. S2 and S3). All models display weak changes in the outgoing LW radiation in the NH until the second post-eruption boreal summer (1992), when a rather sudden drop takes place consistently in all models (Fig. S3). This may reflect a change in the spatial structure of aerosol forcing, with the aerosol cloud predominating over the NH extratropics in the second post-eruption year. Clear-sky net

TOA radiative flux anomalies, calculated as rsdt minus clear-sky outgoing shortwave (rsutcs) minus clear-sky outgoing longwave radiation (rlutcs) showing more significant inter-model differences than full-sky diagnostics for all considered regions (Fig. 4b,d,f,h).

Figure 5 illustrates anomalies of the surface net vertical radiative flux calculated as anomalies of downward shortwave (rsds) plus downward longwave (rlds) minus upward shortwave (rsus) minus upward longwave radiation (rlus). At the global scale,

the models agree substantially as shown by the large overlap between ensemble means and envelopes. The largest reduction in the downward net surface fluxes occurs around the first post-eruption boreal autumn and winter, with average anomalies persisting on values below -2 Wm-2 well into the first post-eruption boreal spring. Two models stand out from the ensemble: MIROC-ES2L and GISS-E2.1-G, the former with weaker and the latter with stronger anomalies during the first two post-

eruption years. As shown for the TOA fluxes, the models agree well in the extratropics while differences are strongest in the tropics. As also seen for the TOA fluxes, changes in the NH are small until the second post-eruption boreal summer.

Figure 6 illustrates anomalies for the surface upward vertical latent plus sensible heat (LH+SH) flux. All models similarly produce a weak global response, identified by small ensemble mean anomalies and large ensemble spread encompassing positive and negative values. However, the models agree on indicating a tendency toward negative upward heat surface flux anomalies during the first three post-eruption years, arguably linked with reduced surface temperatures and ocean heat losses to the atmosphere, and with a slower development of anomalies compared to the radiative fluxes (peak negative values are observed around the second post-eruption boreal summer). Again, the global response is largely determined by the tropics, with ensemble-mean heat flux anomalies in the SH extratropics remaining always around zero, suggesting a very small sensitivity to the forcing and/or small signal-to-noise.

## 4.3 Tropospheric and surface climate response

Figure 7 illustrates near-surface air temperature anomalies. At the global scale, there are only sporadic significant differences in the temperature response with maximum expected cooling ranging across models between about -0.27 °C and -0.38 °C and a multi-model mean of about -0.33 °C. During the post-eruption cooling, the difference between expected responses across models can exceed 0.15 °C, linked to the significantly weaker cooling in MIROC-ES2L compared to other models. Using paired anomalies, the consistency across models at the global scale is very strong in the first two post-eruption years, indicating a progressive cooling until late 1992 when a maximum cooling of about 0.3 °C is attained (Fig. S6). There is no evidence of a significantly weaker cooling in MIROC-ES2L compared to other models in the paired anomalies, which reveals that the response identified in Fig. 7 may reflect biased sampled states in this model (Fig. 2), as further discussed below. Thereafter, the ensemble-mean trajectories depart more from each other also in the paired anomalies, with a quicker recovery for MIROC-ES2L and a slower one for CanESM5 compared to other models. Anomalies remain negative to the end of the simulations, with values between around -0.08 and -0.12 °C in year 1995 in models that extended the integration to this time (IPSL-CM6A-LR, MPI-ESM1.2-LR and CanESM5). Global-mean temperature anomalies from HadCRUT (black lines in Figure 7) are well within the range of simulated anomalies during the whole period and fluctuate around the simulated expected responses until 1995. The peak cooling in late 1992 compares well with expected model responses.

The response in the tropics is seen to be the source of the occasional disagreement in the global-mean temperature across models and reveals model specificities in the cooling phase that do not emerge at the global scale. This is the case for the intermittent significantly colder anomalies seen in MPI-ESM1.2-LR and the warmer conditions of IPSL-CM6A-LR during the second post-eruption boreal summer, which particularly emerge in the paired anomalies. This again suggests a possible effect of biases in the sampled initial conditions in the case of MPI-ESM1.2-LR, considering its slight negative average of sampled ENSO conditions in piControl at the time of peak forcing (Fig. 2) and the fact that paired anomalies do not yield significant differences among models (Fig. S6). Otherwise, this might reflect differences in the applied forcing (see Sect. 4.2) as well as

the consequent activation of certain dynamical responses in only some models. HadCRUT anomalies for the tropics are within the range of simulated responses but generally weaker than expected model responses, particularly from the onset of the eruption to the first half of 1992 and, later, around 1994. In the extra-tropics, surface cooling is consistently stronger in the NH compared to the SH, which can be linked to the larger land cover in the former and to inter-hemispheric asymmetries in the exposure of polar regions to the volcanically induced radiative forcing anomalies. In the NH the response is negligible until the second post-eruption boreal summer, when hemispheric surface temperature anomalies drop until the following boreal winter to reach ensemble-mean values around -0.5 °C in MIROC-ES2L, CanESM5 and MPI-ESM1.2-LR, and around -0.7 °C in GISS-E2.1-G, IPSL-CM6A-LR and UKESM1. HadCRUT anomalies confirm the inter-hemispheric asymmetry in post-eruption cooling. For both hemispheres observations remain largely within the multi-model spread but deviate substantially from expected responses. In the NH this is due to a stronger contribution from internal variability causing large deviations of both observations and individual model realizations from ensemble means especially during boreal winter. In the SH there is a stronger progressive cooling during 1991-1995 in observations compared to expected responses; an explanation is that observed trends pertains to longer-term variability (not shown).

Figure 8 illustrates the precipitation anomalies. The magnitude of the reduction of global-mean precipitation is similar in all models except MIROC-ES2L. Peak ensemble-mean anomalies are smaller than -0.05 mm/day in all models, which is small compared to the ensemble variability of the simulations. The difference between MIROC-ES2L and the other models is reduced if paired anomalies are considered (Fig. S7), which again points to a biased sampling of initial states in this model. The precipitation response is especially small in the extratropics, hence the global reduction of precipitation largely stems from a reduction of the tropical precipitation. Simulated precipitation anomalies compare more poorly with observations than near-surface air temperature anomalies, particularly for the GL and TR regions. For GL, a negative precipitation anomaly around -0.05 mm/day is observed during 1991-1993, hence comparable with expected peak responses, but the evolution of the anomaly before and around the eruption differs between observations and ensemble means, possibly due to an ongoing anomalous dry period in the TR region. There are also several months where observations exceed the multi-model range, both upwards and downwards, suggesting an underestimation of internal variability by the models. As for temperature, models robustly capture the observed inter-hemispheric asymmetry in the post-eruption evolution of precipitation anomalies, with a stronger response in the NH with respect to the SH.

Figure 9 illustrates the response of ENSO and NAO, quantified using the RSST-Nino3.4 index (see Sect. 3) and the box-based NAO index (see Sect. 2.2). RSST-Niño3.4 anomalies indicate a general tendency of ensemble means toward neutral anomalies in the first post-eruption year, except for MIROC-ES2L that shows slight positive anomalies. Then, since early 1992, two clusters of models emerge: The first cluster exhibits the development of warm anomalies in late 1992 followed by negative anomalies in 1993-1994 and includes MIROC-ES2L with ensemble-mean warm anomalies peaking at around 1°C and IPSL-CM6A-LR with ensemble-mean warm anomalies peaking at around 0.5°C; both models diverge in timing and amplitude of the follow-up cold ENSO anomaly. The second cluster exhibits near-neutral ensemble-mean anomalies throughout the simulation period and includes GISS-E2.1-G (showing a slight warm ENSO anomaly in 1992 in the ensemble-mean), MPI-

ESM1.2-LR, CanESM5 and UKESM1. MIROC-ES2L stands out as significantly different from the other models both concerning the warming in 1992 and the following cooling around 1994. Considering the possible biased sampling of ENSO states, Figs. 9c and S8 illustrate paired anomalies of RSST-Niño3.4 for the volc-pinatubo-full simulations and corresponding piControl sections, and the anomalies for such piControl sections from the climatology, respectively. Results indicate consistency across model responses in the early post-eruption phase (until early 1992), associated with a generally weak

response, seen as ensemble-mean trajectories remaining within the ±0.25 °C range with the noticeable exceptions of GISS-E2.1-G, showing a clearer initial tendency toward negative RSST-Nino3.4 anomalies compared to other models, and IPSL-CM6A-LR, for which there is remarkable consistency between the ensemble-mean paired anomalies and anomalies with respect to the climatology showing a warm ENSO response in early 1992. In the second post-eruption year, paired anomalies indicate that models cluster into two main groups, but with some noticeable differences with respect to indications from

anomalies with respect to the climatology. There is still a clear tendency toward warming in MIROC-ES2L during the second post-eruption winter (Fig. 9c), which peaks at much lower values compared to the anomalies with respect to the climatology (Fig. 9). Anomalies with respect to the climatology for the piControl sections confirm that part of the response detected in MIROC-ES2L is in fact spurious and linked to a biased sampling of warm ENSO states in piControl (Fig. S8). Paired anomalies also show a stronger warming in 1992 and 1993 for GISS-E2.1-G compared to the anomalies from the climatology, and a

tendency in CanESM5 toward a prolonged cold ENSO response in late 1992 and 1993. Overall, the models thus seem to agree on a near-neutral (or no) ENSO response in the early phase, whereas the models disagree on the later response of ENSO, with some models suggesting a warm (El Niño-like) response and others suggesting a neutral (or no) response or even a cold (La Niña-like) response. The different sign and timing of the response highlight the potential influence of the different simulation of ENSO dynamics in the different models. In addition to the sampling bias, the large ensemble spread indicates a general low

signal-to-noise ratio.

    The NAO response and inter-model agreement are also difficult to interpret based on the chosen diagnostic, due to the apparent low signal-to-noise ratio of the response. There is a weak tendency toward positive NAO anomalies in the first post-eruption winter in GISS-E2.1-G and IPSL-CM6A-LR, and earlier in UKESM1, which contrasts with tendential negative anomalies in CanESM5.

**4.4 Feedbacks**

    Figure 10 illustrates diagnostics that relate to two examples of climate feedbacks. The first describes changes in the atmospheric LW transmittance through the LWt/LWs↑ ratio, defined as the ratio between the value of the LWt/LWs↑ ratio calculated for the volc-pinatubo-full simulations and for the corresponding piControl sections. This value integrates the effects of diverse processes, including absorption by the volcanic aerosol and high-level clouds, and several feedbacks including Planck, lapse

rate and water vapor. The second describes changes in solar radiation linked to the cloud-albedo feedback, estimated as the ratio between the value of the SWt/SWtcs ratio calculated for the volc-pinatubo-full simulations and for the corresponding

piControl sections. There is a tendency in all models to yield a smaller LWt/LWs↑ ratio under volcanically perturbed compared to unperturbed conditions, but possibly less strong for IPSL-CM6A-LR, which has a colder mean state of the global surface climate, and strongest for UKESM1, which has a warmer mean state of the global surface climate than all other models except

MIROC-ES2L. The inter-model difference between models agrees with the general relation that a warmer climate has a stronger water vapor feedback, which is accounted for in this diagnostic, but other factors influence the thermal radiation response across the atmosphere, possibly including aerosol radiative effects. However, there seems to be no major difference between the first and the second post-eruption year in terms of LWt/LWs↑. Given the substantial difference in aerosol loading between both years, this again suggests that the diagnostics mostly reflect differences in feedbacks operating through changes

in the LW radiation.

There is a clear strengthening of the cloud-albedo feedback in all models, as the associated diagnostic is largely below the value of 1 in all models, meaning that SWt/SWtcs is smaller under volcanically perturbed compared to unperturbed conditions. There seems to be a clustering between models, with MIROC-ES2L and GISS-E2.1-G with a strong response of the feedback during the first post-eruption year and a strong recovery in the second post-eruption year, IPSL-CM6A-LR with a

comparatively weak response in the first post-eruption year but a stronger persistence of the signal in the second post-eruption year, and CanESM5, UKESM1 and MPI-ESM1.2-LR with an intermediate behavior. The warmer mean state of tropical temperatures of GISS-E2.1-G and MIROC-ES2L compared to other models (Fig. 3) suggest a dependency on temperature, although the clustering may reflect different choices in the parameterization of clouds in the different models as UKESM1, with the warmest climatological tropical temperatures, and CanESM5, with similar climatological temperatures to MIROC-

ES2L, show weaker cloud responses. The diagnostics may also reflect differences in the applied forcing, although the linkage is nontrivial as, for instance, IPSL-CM6A-LR and UKESM1 yield similar variations in rsut/rsutcs under volcanic forcing with respect to unperturbed conditions but have different forcing (compare with Fig. 4).

### 4.5 Effect of sampling strategy

Figure 11 illustrates the effect of the sampling strategy, i.e., the considered ENSO and NAO conditions to start the volc-

pinatubo-full simulations from the piControl following the VolMIP protocol, for the post-eruption climate anomalies. The figure compares empirical distributions for near-surface air temperature anomalies in terms of seasonal-average anomalies for the 1992 boreal summer grouped by different states of ENSO and NAO, sampled using the associated indices defined by the VolMIP protocol (see Sects. 2.1 and 2.8). For global-average near-surface air temperature (see associated data in Table S2), all models show smaller negative temperature anomalies on average in the realizations starting from ENSO+ pre-conditions,

and stronger negative temperature anomalies on average for realizations starting from ENSO- pre-conditions. This result is unsurprising and can be explained by the global temperature anomaly resulting from the ENSO state at the time of maximum cooling superimposing on the volcanic cooling. Note, however, that the anomalies contain the potential effect of sampling biases regarding ENSO, most importantly regarding MIROC-ES2L (Figs. 2 and S9): unbalanced sampling of ENSO can thus

lead to biases in the global-average post-eruption anomalies for some models (see also, e.g., Lehner et al., 2016). This hypothesis is supported by analysis of paired anomalies (Fig. S9) showing no substantial change in the response as a function of ENSO pre-conditioning, or even opposite dependencies of the response on ENSO compared to what is seen in the anomaly analysis (compare Fig. 11 and Fig. S9 for, e.g., GISS-E2.1-G). In any case, the distributions of global-mean temperature anomalies for the different ENSO pre-conditions do overlap considerably in most cases also in the anomaly analysis, indicating that the effect of ENSO preconditioning can be overwhelmed by other factors contributing to internal variability. In particular, the distributions for IPSL-CM6A-LR overlap considerably, suggesting that in this model global cooling is weakly sensitive to the ENSO pre-conditioning. The Nino3.4 index defined as reference for ENSO in the VolMIP protocol potentially contains tropical SST signals that do not pertain to ENSO. A comparative assessment of results obtained using the RSST-Nino3.4 index as grouping criterion does not yield significant differences compared to the original analysis (Figure S10). This is confirmed also for the first post-eruption winter (Figure S11). Finally, a dependency on ENSO preconditioning is found also for the global-mean precipitation response in the second-post eruption summer (Supplementary Table S3).

The NAO sampling affects the global response with an overall weaker impact compared to ENSO, with only some models showing differences in the ensemble-mean response under different NAO pre-conditioning. This can be understood with the weaker imprint of NAO on the global surface climate compared to ENSO, particularly in summer.

Similar considerations stand for regional cooling over areas deemed most impacted by the two considered modes, i.e., the tropics for ENSO and the Northern Hemisphere for the NAO. Preconditioning of ENSO clearly impacts tropical temperatures in the anomaly analysis, with all models agreeing on a weaker cooling under El-Niño preconditioning compared to neutral or La-Niña preconditioning. Again, paired anomalies weaken the dependency of the response to the state of ENSO, revealing that the evolution of post-eruption tropical temperature anomalies contain the signal of dynamics related to ENSO and unaffected by the forcing. The lack of impact of NAO sampling on NH temperatures can be again explained by the fact that the NAO is predominant in winter whereas direct radiative responses are better identified in the summer season. Further, the NAO hemispheric pattern is strongly heterogeneous and includes both warm and cold regional temperature anomalies within the Northern Hemisphere that tend to compensate for each other leaving a negligible imprint on hemispheric averages.

### 4.6 Ensemble size and spread

Figure 12 illustrates the effect of sample size on the uncertainty related to the expected (i.e., ensemble mean) surface temperature and precipitation response, shown in terms of standard error of the mean of post-eruption anomalies calculated for 1992 annual averages (see Sect. 2.2). The standard errors converge toward the value obtained for the 25-member ensemble in all models, starting from the higher and more uncertain estimates obtained for low ensemble sizes. Otherwise, the curves differ across models indicating that they disagree on how the ensemble size affects the standard error of the ensemble mean. Models rank similarly concerning errors in global-mean temperature and global-mean precipitation, reflecting similar relative uncertainty in the response of both variables.

For small ensemble sizes, the amplitude of the 5-95 percentile range of the standard error varies substantially across models, with larger values in IPSL-CM6A-LR, GISS-E2.1-G, MIROC-ES2L and UKESM1 compared to MPI-ESM1.2-LR and CanESM5. This reflects a weak signal-to-noise ratio of the response in the former group of models as seen in their larger full-size standard errors and highlights the exposure of these models to potentially large sampling biases in the expected response

when it is estimated from a few events. Overall, uncertainty in the ensemble mean strongly depends on both, ensemble size and model, which, together with the variety of unperturbed climatologies expressed by the models in the piControl, prevents generalization and requires model-specific assessments of the signal-to-noise ratio of post-eruption anomalies.

## 5 Discussion

In the following, we illustrate major gaps of knowledge emerging from our analyses to be addressed in follow up studies (Sect.

5.1) and discuss possible improvements to the experimental design and the protocol of VolMIP in light of a possible second phase of the initiative (Sect. 5.2).

### 5.1 Gaps of knowledge

Overall, the volc-pinatubo-full results indicate a general agreement in the surface climate response to volcanic forcing among different models compared to previous results. The VolMIP protocol allows models to be compared more powerfully by

sampling across different states of dominant climatic modes. This contrasts with the small yet significant inter-model differences in volcanic aerosol optical depth and post-eruption radiative flux anomalies that call for more in-depth analysis of the volc-pinatubo-full simulations and question the efficiency of the VolMIP protocol for constraints on the forcing data across models.

The apparent differences in the aerosol forcing implemented in the different models require further work to be fully assessed

and understood. They may reflect differences in model physics, including radiative schemes and parameterizations of aerosol-SW and -LW interactions. We recommend first checking the details of the stratospheric aerosol optical, single scattering albedo and asymmetry parameter depth diagnosed for each model and identifying model specificities regarding the tropopause height, especially over the tropics, and choices in the replacement of aerosol data below the tropopause. Inter-model differences in radiative flux anomalies are seen both at the top-of-atmosphere in full-sky and especially clear-sky diagnostics, and at the

surface. They may also indicate inter-model differences in model radiative codes that rely on different spectral band resolution and schemes for aerosol-radiation interactions, or in adjustments/feedbacks, for example cloud adjustments (Schmidt et al., 2018), or the global water balance (Wild, 2020). We recommend analysis of effective radiative forcing or instantaneous radiative forcing calculations (e.g., Smith et al., 2018). In this regard, the VolMIP protocol has defined a group of variables to diagnose volcanic instantaneous radiative forcing (Table 4 in Zanchettin et al., 2016), which were requested to generate

volcanic forcing for the volc-pinatubo-surf/strat experiments and can be useful to better constrain the imposed aerosol forcing in the different models.

The idealized nature of the VolMIP experiments does not allow a direct comparison with observations, which must rely on output from full-forcing transient simulations. In this regard, analysis of CMIP6-historical simulations (Pauling et al., 2021) provide a much better agreement with observations compared to CMIP5-historical simulations concerning the global-mean surface temperature response to the 1991 Pinatubo eruption. The preliminary results shown here confirm that the observed post-Pinatubo anomalies agree well with the expected simulated response to a Pinatubo-like eruption, especially as far as the global-mean surface temperature is concerned. The post-eruption temperature evolutions over smaller regions are increasingly affected by internal variability, as different regional mechanisms/feedbacks may be activated under different mean background states and yield expected responses that are consistent at the global scale while differing at regional scales (e.g., Zanchettin et al., 2013). A comparative assessment of inter-model consistency in the climate response to the 1991 Mt. Pinatubo eruption in CMIP6-historical and in the volc-pinatubo-full experiment could also help to clarify the impact of boundary conditions and choices regarding the correction to the volcanic aerosol input data to confine volcanic aerosol to the stratosphere for volc-pinatubo-full.

The tropics emerge as a key region to understand inter-model differences in the volc-pinatubo-full ensemble and assess the realism of the simulated climate response to volcanic forcing. Fiedler et al. (2020) analysed the simulated tropical precipitation across different phases of CMIP and found similar behaviors for CMIP5 and CMIP6 models. In both cases the expected post-eruption reduction in precipitation over land is stronger than what is indicated by observations. This suggests a too-strong response of tropical precipitation to volcanic aerosols persisting across different model generations. Our preliminary results on total (ocean plus land) precipitation are rather inconclusive regarding the realism of the simulated responses in the tropics and at the global scale, due to the strong internal variability expressed by precipitation in both models and especially observations. However, an explanation based on CMIP5 results indicates that post-eruption precipitation anomalies strongly depend on both the magnitude of applied volcanic forcing and the state of the ocean at the time of eruption (Paik et al., 2020). Our results confirm the strong dependence of the precipitation response - and more generally of the climate response - to both, the mean climate state, and the phase of internal climate variability at the time of eruption, beyond the obvious considerations about differences in the magnitude of the applied forcing discussed above. This was highlighted here especially for the case of the biased sampling of ENSO conditions in one of the contributing models (MIROC-ES2L), which reverberated on global-scale responses of temperature and precipitation. In fact, despite our results suggesting that the implementation of the experiment protocol was overall effective for most of the contributing models, room for possible improvements is evident, especially the strictness of the sampling of initial conditions for the volc-pinatubo simulations.

The dependency of post-eruption anomalies on initial conditions emerges as one of the clearest results of our analysis. Based on our analysis, ENSO does not show a robust response neither across models nor within individual models, possibly except for IPSL-CM6A-LR that yields a clear warm ENSO response in the second post-eruption winter. This implies that ENSO evolution determined by ongoing intrinsic dynamics can significantly affect post-eruption anomalies at the global, hemispheric, and regional scales. Therefore, our results highlight how for an eruption like the 1991 Mt. Pinatubo a biased sampling of internal variability may lead to non-negligible biases in the estimation of the expected climate response and call for caution in

the assessment of post-eruption anomalies. Initial conditions have the potential to affect comparability between simulations and observations. An analysis of the temperature response in sub-ensembles including realizations with a neutral pre-eruption state of ENSO like observations (Figure S12) illustrates how sampling of initial conditions may affect model-data comparability as in-phase internal variability emerges in the ensemble-mean. Results point to an overall better representation

of the observed peak global cooling in the neutral ENSO sub-ensemble, but new model-specificities arise as well (e.g., a faster post-eruption cooling in UKESM1). Sampling of neutral pre-eruption states of ENSO as in observations yields only negligible changes in the tropical precipitation response in the volc-pinatubo-full simulations compared to the full-ensemble analysis (Figure S13). Especially, no model robustly simulates the reduced precipitation observed in the early post-Pinatubo phase. It is arguable that a neutral ENSO pre-eruption state as diagnosed here based on the Nino3.4 index for the winter months

preceding the eruption only weakly constrains ENSO tendencies and, more generally, the climate evolution in the following year. This possibility concerns already intrinsic dynamics due to the spring predictability barrier for ENSO (e.g., Jin et al., 2008). Conclusions in this regard require additional analyses, which must rely on a more complete set of ENSO diagnostics than that used here. The choice of indices describing the state of ENSO appears potentially relevant for the assessment of dynamical responses as well, including the response of ENSO itself. Despite the broad scattering and large overlap of values

of ENSO and NAO at the time of peak forcing under unperturbed and perturbed states (Fig. 9e) suggesting a lack of robust response of both modes to volcanic forcing, there are known limitations in the considered indices and further studies shall consider improved diagnostics for both modes of climate variability. For ENSO, the models used here exhibit different relations between SSTs in the central equatorial Pacific and the tropical average SST as seen by the variable relation between Nino3.4 and RSST-Nino3.4 indices. Such differences concern the piControl simulations, hence intrinsic simulated dynamics,

and may reflect a dependency of ENSO characteristics on tropical SST biases, which remains to be understood. For the NAO, detection of volcanic signals could be improved by analyses focused on the winter season, when the mode is most stable, and in association with the more hemispheric-scale Arctic Oscillation.

Whereas dependency of the simulated responses on initial conditions and background climate state, including presence and magnitude of additional forcing factors, thus remains to be fully explored in follow-up studies, this analysis already shows that

the use of paired anomaly calculations mitigates the effect of sampling biases. The role of initial conditions in shaping the climate response to larger magnitude eruptions has been subject of recent studies (e.g., Zanchettin et al., 2019; Pausata et al., 2020). Analysis of the output of the VolMIP volc-long-eq experiment, based on idealized climate simulations of the 1815 Mt. Tambora eruption, will provide context to the general conclusions drawn here for the 1991 Mt. Pinatubo eruption. In addition, the VolMIP Tier 3 volc-pinatubo-slab experiment can provide very useful insights: it uses the same forcing as volc-pinatubo-

full but a slab ocean in order to clarify the role of coupled atmosphere–ocean processes for the dynamical response of ENSO (Zanchettin et al., 2016).

Accounting for sampling of initial conditions is relevant when investigating other known dynamical responses, for instance the post-eruption Northern Hemisphere winter warming. Coupe and Robock (2021) found that if the observed sea-surface temperatures are prescribed, the NCAR Community Earth System Model, with the Community Atmospheric Model 5,

realistically simulates the observed winter warming after the three largest volcanic eruptions of the late 20th Century, but it fails if the ocean model is coupled to the atmosphere. We foster investigation of the post-eruption winter warming simulated by the volc-pinatubo-full ensemble, and recommend that results are interpreted accounting for the state of ocean variability in each simulation and also for climatological biases/differences in ocean-atmosphere coupled processes. Depending on the scientific question, Atmosphere Model Intercomparison Project (AMIP) style experiments with prescribed sea-surface temperatures might be an alternative approach to coupled climate experiments.

## 5.2 Implications for VolMIP

Recent advances in the design of climate model experiments makes some afterthoughts necessary regarding the VolMIP protocol, including the sampling strategy. The identification of specific conditions of ENSO and NAO (or of any other relevant climatic mode) to start the volc-pinatubo simulations from the piControl might seem not necessary in light of the prospect to increase the integration and assessment of large ensemble experiments within the next phase of CMIP (Deser et al., 2020). If the way forward is toward the so-called "single model initial-condition large ensembles" (SMILEs), discussion about supervised sampling strategies may appear obsolete: SMILEs could provide many realizations of historical eruptions, including the 1991 Mt. Pinatubo, with good sampling of initial conditions as part of the DECK-historical simulations. However, the lack in transient simulations of unperturbed climate evolutions corresponding to periods during and after volcanic forcing would impede fully disentangling forced and intrinsic components of climate evolutions, as evidenced here for some relevant aspects of climate variability including ENSO. In this sense, idealized experiments as those originally proposed for VolMIP remain a valuable contribution to understand the climate response to volcanic forcing and its simulation.

Another promising approach for the future is also the application and combination of different SMILEs. Maher et al. (2021) demonstrated the utility of combining different types of SMILEs to identify which part of post-eruption climate evolution is a response forced by the volcanic eruption and which one is due to other sources. The combination of different types of SMILEs might be a potential way to move forward to answer open scientific questions, such as the causes of post-eruption winter warming or post-eruption tropical sea-surface temperature variability, by separating and quantifying the forced response from internal variability on a regional scale.

Pauling et al. (2021) identify no robust connection between ECS and the post-Pinatubo global cooling in an ensemble of CMIP6 historical simulations. Figure 13 illustrates how ECS relates to seasonal-average near-surface air temperature anomalies in the volc-pinatubo-full ensemble. The results overall agree with the conclusion by Pauling et al. (2021) that ECS does not play an important role for the global-mean temperature response to a Pinatubo-like eruption, not in the expected (average) response nor in its uncertainty. Again, MIROC-ES2L stands out from the other models, with smaller post-eruption cooling than observed for MPI-ESM1.2-LR, which has similar ECS. This difference is much reduced for calculations based on paired anomalies, with inter-quartile ranges of both models overlapping in the case of global-mean temperature anomalies for the second post-eruption boreal summer (not shown). It will be important to investigate this relation for the case of a stronger eruption, based on the volc-long experiments.

If activities are continued in a second phase of VolMIP with a Pinatubo-like set of experiments analogous to volc-pinatubo, we make the following considerations and propose the following improvements to the protocol.

Concerning the forcing, the original VolMIP core experiments focused on two historical tropical eruptions (Pinatubo, Tambora) with hemispherically symmetric forcing. However, the transport of aerosol from the tropics to each hemisphere is known to be quite variable for tropical eruptions depending on the eruption latitude and season. While the 1991 Mt. Pinatubo eruption produced a volcanic aerosol cloud that spread relatively evenly in the Northern and Southern Hemispheres, the volcanic aerosol distribution after the 1982 El Chichón eruption and the 1963 Agung eruption were heavily biased to one

hemisphere. Previous studies on tropical eruptions already have pointed out the importance of asymmetric volcanic forcing on tropical rain belts or cyclone activities (e.g. Yang et al., 2019; Jacobsen et al., 2020) or for the comparison with proxy data (e.g. Timmreck et al., 2021). Therefore, while we support the vision that VolMIP must remain an idealized volcanic forcing experiment, improvement in the direction of accounting for inter-hemispheric forcing asymmetries should be discussed. In this regard, the original VolMIP protocol included two experiments with strongly asymmetric eruptions, namely volc-long-hlN

(high-latitude eruption in the Northern Hemisphere) and volc-long-hlS (high-latitude eruption in the Southern Hemisphere). These experiments, currently set at priority levels Tier 2 and Tier 3, respectively, may provide valuable information as endmembers in an ensemble of idealized volcanic forcing experiments tagging uncertainties due to the spatial structure of the aerosol forcing.

Concerning ensemble size, length and sampling of initial conditions, the current recommended minimum ensemble size (25)

seems to be sufficient whereas a longer integration time is proposed (minimum 5 years). Nonetheless, already in the current phase of VolMIP a few contributing modelling groups generated a much larger ensemble. In the future, a balance must be established between the use of SMILEs and volcanic simulations with controlled selection of initial conditions from a control simulation. For the latter, we foresee a shift of focus from the radiative response to dynamical responses. Accordingly, we recommend shifting the focus only on ENSO for the sampling of initial conditions, since the NAO seems to have an only

limited impact on the response and could therefore be neglected. We suggest the protocol be updated so that the ENSO mean state and tendency on the period from the last pre-eruption winter to the onset of the eruption is considered, instead of the state during the first post-eruption winter as in the original VolMIP protocol. Instead of indices based on sea-surface temperatures, we recommend using diagnostics that more closely tie to processes relevant for ENSO dynamics, for instance the equatorial Pacific ocean heat content described by indices such as the Warm Water Volume index (Meinen and McPhaden, 2000). This

will allow identification of how ENSO preconditioning affects ENSO's response to the eruption and the role of the state of the equatorial Pacific at the time of eruption for the broad climatic response to be disentangled. More generally, given the variable representation of ENSO - and other climatic modes as well - in different climate models, the choice of the associated index should reflect physical understanding of the climate mode rather than merely build on a mathematical construct. For the NAO, at least, a recent multi-model study using the same index definition employed in volc-pinatubo-full suggests a marked

consistency across CMIP6 models (Cusinato et al., 2021). In the future, the choice of an index should be supported as much as possible by preliminary assessments of climate model biases.

Additional modes of climate variability may be considered, which can be identified based on their relevance for the response in follow-up composite analyses. Among potentially relevant modes, the Quasi Biennial Oscillation (QBO) (e.g., Thomas et al., 2009) did not have explicit focus in VolMIP, but it is arguable that its representation in climate models will continue

improving. However, the prescribed aerosol optical properties at the basis of VolMIP constitute a major limitation to an effective implementation of a sampling strategy for the QBO, since the phase of the QBO affects stratospheric transport including that of stratospheric aerosol, hence ultimately the volcanic forcing (e.g., Hommel et al., 2015). The effects of inconsistencies between QBO and prescribed forcing on volc-pinatubo experiments are unknown. Until this gap of knowledge is filled, we recommend continuing to sample an easterly phase of the QBO at the time of the eruption whenever possible.

The phase of modes of variability with longer characteristic timescales may be important as well. For instance, Illing et al. (2018) identify significant regional differences in near-surface air temperature over the North Atlantic, sea-ice area fraction, frost days, and precipitation between two Pinatubo-like experiments, which were initialized in years with different phases of the Pacific Decadal Oscillation. Then, the state of the North Atlantic ocean circulation as described by the Atlantic Multidecadal Variability (AMV) may affect atmospheric responses to the volcanic eruption as well (e.g., Omrani et al., 2016;

Ménégoz et al., 2018; Coupe and Robock, 2021). Decomposition of the AMV signal in paleoclimate simulations also suggests that the internal component of the AMV, which is tightly connected with the Atlantic meridional overturning circulation, lacks robust behaviour across simulations during periods of major volcanic forcing (Fang et al., 2021). VolMIP experiments are well suited to test such hypothesis. Therefore, the possibility to include AMV and Pacific Decadal Oscillation in the sampling protocol should be considered, either with a strict explicit definition of their phase or with its a posteriori assessment in case

of response biases across models. Then, we advocate the development and sharing of an algorithm for sampling initial conditions to ensure consistency across participating models.

The usefulness of the volc-pinatubo-full experiment cannot be fully understood unless in connection with the companion volc-pinatubo-surf/strat experiments, which are also Tier 1 VolMIP experiments and were designed to disentangle dynamical responses to the two primary thermodynamic consequences of aerosol forcing, i.e., surface cooling and stratospheric heating.

Analysis of these experiments will allow us to clarify the main pathways through which volcanic aerosols affect atmospheric circulation and surface climates. This type of mechanistic experiments might be useful also for new questions to be addressed in a potential second phase of VolMIP that focus on the impact of volcanic aerosol on stratospheric/atmospheric dynamics and chemistry.

As final considerations, as uncertainties generated by aerosol chemical and microphysical properties are neglected in VolMIP,

it is crucial for advancing understanding and prediction of the climatic response to volcanic eruptions that VolMIP activities continue to be interlinked with activities within the SPARC/SSiRC Interactive Stratospheric Aerosol Model Intercomparison project (ISA-MIP, Timmreck et al., 2018). This is necessary, for instance, to pave the way for an explicit consideration of the QBO in VolMIP experiments. Then, a general critical aspect about VolMIP is the long turn over time between the experiment design, the integration of the simulations, and the analysis of the output. When most participating modeling groups performed

the volc-pinatubo-full simulations, the experiment protocol was over five years old. Hence, some of the questions raised in the

VolMIP overview paper in 2016 that steered the set-up of VolMIP experiments have been answered in the meantime, while new questions have arisen. Still, we have outlined the potential for VolMIP to contribute to answering these emergent new questions, thanks to its well-designed experimental protocol and, especially to the international community that has been built around the initiative.

## 6 Conclusions

First results from the VolMIP volc-pinatubo-full experiment reveal a dichotomy in the simulated climate response to a Pinatubo-like eruption, which is seen as broad inter-model consistency of post-eruption surface climate anomalies contrasting with small yet significant differences in post-eruption radiative flux anomalies. Despite further analysis of volc-pinatubo-full needed to explain such inter-model differences, the preliminary results shown here indicate that they reflect differences in the applied forcing. As well-constrained volcanic forcing is a pillar of VolMIP, any ambiguity in the protocol - possibly the treatment of volcanic aerosol input data at and below the tropopause - shall be amended in a possible second phase of the initiative. Then, the statistical consistency diagnosed in the near-surface air temperature and precipitation response may simply reflect the large intrinsic variability of the associated processes compensating for forcing uncertainties, which is also seen in single-model analyses as dependency of the response on the climate state at the time of eruption. Improved assessment of initial condition influences on direct radiative and dynamical responses is therefore also recommended towards a refinement of the volc-pinatubo sampling protocol.

**Code and data availability**

The raw data used in this study are part of the output of CMIP6. The CMIP6 data are available at https://esgf-data.dkrz.de/projects/esgf-dkrz/. The time series used in the analysis are available from the World Data Center for Climate (WDCC) public repository at the Deutsches Klimarechenzentrum (DKRZ) (Zanchettin et al., 2022a,b). All models used in this study are official release versions for CMIP6. Table 1 provides the doi of the output descriptions for all models, which also include information about their version number as well as that of their submodels. The code of CanESM5 is available at https://doi.org/10.5281/zenodo.3251113. The code of "version E2.1 of Model E" of GISS-E2.1-G is available at https://simplex.giss.nasa.gov/snapshots/. The code of MPI-ESM1.2-LR is available to the scientific community under licenses, which define the conditions under which the models, component models, and other software can be accessed and used, see https://mpimet.mpg.de/en/science/modeling-with-icon/code-availability for further information. The code of MIROC-ES2L is available at https://doi.org/10.5281/zenodo.5975701. The IPSL-CM6A-LR model code used in this work was frozen (Version 6.1.0) and subsequently altered only for correcting diagnostics or allowing further options and configurations. Versions 6.1.0 to 6.1.11 are therefore bit-reproducible for a given domain decomposition, compiling options, and supercomputer. LMDZ, XIOS, NEMO, and ORCHIDEE are released under the terms of the CeCILL license. OASIS-MCT is released under the terms

of the Lesser GNU General Public License (LGPL). IPSL-CM6A-LR code (Version 6.1.0) is publicly available through svn, with the following command lines: svn co https://forge.ipsl.jussieu.fr/igcmg/browser/modipsl/branches/publications/IPSLCM6.1.11-LR_05012021 modipsl cd modipsl/util; ./model IPSLCM6.1.11-LR. The mod.def file provides information regarding the different revisions used,

namely: (1) NEMOGCM branch nemov36STABLE revision 9455; (2) XIOS2 branchs/xios-2.5 revision 1873; (3) IOIPSL/src svn tags/v224; (4) LMDZ6 branches/IPSLCM6.0.15 rev 3643; (5) tags/ORCHIDEE20/ORCHIDEE revision 6592; (6) OASIS3-MCT 2.0branch (rev 4775 IPSL server). The login/password combination requested at first use to download the ORCHIDEE component is anonymous/anonymous. We recommend referring to the project website: http://forge.ipsl.jussieu.fr/igcmg_doc/wiki/Doc/Config/IPSLCM6 for a proper installation and compilation of the environment

(version 6.1.10). The UK Earth System model is documented in Sellar et al (2019a). Due to intellectual property right restrictions, we cannot provide the source code or documentation papers for the UM. The Met Office Unified Model is available for use under licence. A number of research organizations and national meteorological services use the UM in collaboration with the Met Office to undertake basic atmospheric process research, produce forecasts, develop the UM code, and build and evaluate Earth system models. For further information on how to apply

for a licence, see http://www.metoffice.gov.uk/research/modelling-systems/unified-model

**Competing interests**

The authors declare no competing interest.

**Acknowledgments**

Resources supporting this work were provided by the NASA High-End Computing (HEC) Program through the NASA Center
for Climate Simulation (NCCS) at Goddard Space Flight Center. The simulations by MIROC-ES2L used the Earth Simulator and JAMSTEC Super Computing System, supported by TOUGOU/SOUSEI, the Integrated Research Program for Advancing Climate Models (grant number JPMXD0717935715)/Program for Risk Information on Climate Change, through the Ministry of Education, Culture, Sports, Science, and Technology of Japan. MPI-ESM1.2-LR simulations were performed at the Deutsches Klimarechenzentrum (DKRZ). The IPSL-CM6A-LR experiments were performed using the HPC resources of
TGCC under the allocations 2019-A0060107732, 2020-A0080107732 and 2021-A0100107732 (project gencmip6) provided by GENCI (Grand Equipement National de Calcul Intensif). As such it benefited from the French state aid managed by the ANR under the "Investissements d'avenir" program with the reference ANR-11-IDEX-0004-17-EURE-0006. This study benefited from the ESPRI (Ensemble de Services Pour la Recherche l'IPSL) computing and data center (https://mesocentre.ipsl.fr) which is supported by CNRS, Sorbonne Université, École Polytechnique, and CNES and through
national and international grants. The UKESM simulations were carried out on the joint UK Met Office and Natural

Environment Research Council (NERC) MONSooN supercomputing system ("Met Office and NERC Supercomputer Nodes"), with data analysis using the UK collaborative "Joint Analysis System Meeting Infrastructure Needs" super-data-cluster facility (JASMIN).

**Financial support**

This research has been supported by the Deutsche Forschungsgemeinschaft Research Unit VolImpact (FOR2820, grant no. 398006378, CT) within the project VolClim, and the German Federal Ministry of Education and Research (BMBF) within the research programme "ROMIC-II, ISOVIC" (FKZ: 01LG1909B, SWF). The simulation by MIROC-ES2L has been supported by the Ministry of Education, Culture, Sports, Science and Technology (MEXT) of Japan (Integrated Research Program for Advancing Climate Models, grant no. JPMXD0717935715). This work was undertaken in the framework of the French L-

IPSL LABEX and the IPSL Climate Graduate School EUR and benefited from the FNS "SYNERGIA "EffeCts of lArge voLcanic eruptions on climate and societies: UnDerstanding impacts of past Events and related subsidence cRises to evAluate potential risks in the future"(CALDERA) project under French CNRS Grant Agreement number CRSII5_183571 - CALDERA. Climate modeling at GISS is supported by the NASA Modeling, Analysis and Prediction program. AR is supported by US National Science Foundation grant AGS-2017113. BJ was funded under the joint UK BEIS/DEFRA–Met

Office Hadley Centre Climate Programme (GA01101). GWM was funded via the UK National Centre for Atmospheric Science (NCAS), via the ACSIS long-term science programme on the North Atlantic climate system (NE/N018001/1). WF acknowledges funding from the NCAS single-centre Long-term Science programme (NE/R015244/1), within the "Climate and High Impact Weather" theme. HW was supported under REU Grant number OCE 17-57602 as part of the 2019 Lamont-Doherty Earth Observatory Summer REU program.

**Author contributions**

DZ, CT, and MK designed the study. DZ performed the statistical analyses, drafted the initial manuscript and coordinated the writing process. CT, MK, AS and MT contributed to writing the paper. CT configured the MPI-ESM1.2-LR simulations and provided the output data with the help of SWF; MK and NL performed the IPSL-CM6A-LR simulations and provided the output data with the help of SB; JC and LR run the CanESM5 simulations and provided the output data; KT run the GISS-

E2.1-G simulations and provided the output data with the help of HW and ANL; MA run the MIROC-ES2L simulations and provided the output data; WF and GWM co-configured the UKESM1 volc-pinatubo-full ensemble, with WF running the 54 simulations, GM extracting and post-processing the data for this analysis. BJ generated the CMIP6 volcanic forcing ancillary files for UKESM1. All authors contributed to discussion and finalization of the manuscript.

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

| Model | Institute | Atm. Res. | Ocean Res. | QBO | Ozone | ECS (K) | Reference | Output data description |
|---|---|---|---|---|---|---|---|---|
| CanESM5 | Canadian Centre for Climate Modelling and Analysis | ~2.8°x2.8° 49 levels up to 1 hPa | 1°x1°; 45 levels | | Prescribed | 5.6 | (Swart et al., 2019) | doi:10.22033/ ESGF/CMIP6 .1303 |
| GISS-E2.1-G | NASA Goddard Institute for Space Studies | 2°x2.5°; 40 levels up to 0.01 hPa | 1°x1.25°; 40 levels | No, easterly phase dominates | Prescribed, based on prognostic O3 simulations with the same model version (CMIP6 physics version 3) | 3.59 | (Kelley et al., 2020) | doi:10.22033/ ESGF/CMIP6 .1400 |
| IPSL-CM6A-LR | Institut Pierre-Simon Laplace | 1.25°x2.5°; 79 vertical levels up to 80km | 1° nominal resolution with refinement of 1/3° in the equatorial region; 75 | yes | Prescribed | 5.01 | Boucher et al. (2020) | doi:10.22033/ ESGF/CMIP6 .1534 |

| | | | vertical levels | | | | | |
|---|---|---|---|---|---|---|---|---|
| MIROC-ES2L | Japan Agency for Marine-Earth Science and Technology/ Atmosphere and Ocean Research Institute, The University of Tokyo/ National Institute for environmental Studies | ~2.8°x2.8°; 40 levels up to 3 hPa | >63ºN: ~60 km (zonal) x ~33 km (meridional), <63ºN: 1º (long.) x 0.5-1º (lat.) , 62 vertical levels | no | Prescribed | 2.7 | (Hajima et al., 2020) | doi:10.22033/ ESGF/CMIP6 .902 |
| MPI-ESM1.2-LR | MPI für Meteorologie | 1.9°x1.9°, 47 lev, up to 0.01 hPa, 13 lev. above 100 hPa | GR15, 40 levels | no | prescribed | 2.83 | (Mauritsen et al., 2019) | doi:10.22033/ ESGF/CMIP6 .742 |
| UKESM1 | UK Met Office and UK Natural Environment Research Council (NERC). | 1.875º longitude x 1.25º latitude (~135 km horiz) with 85 vertical levels (hybrid-height) to 85km (50 of | 1ºx1º horizontal resolution with 75 vertical levels | yes | Interactive via stratosphere-troposphere chemistry (Archibald et al., 2020) | 4.7 | (Sellar et al., 2019) | doi:10.22033/ ESGF/CMIP6 .1569 |

| | | 85    levels below 18 km). | | | | | |
|---|---|---|---|---|---|---|---|

**Table 1: Characteristics of the models participating in the volc-pinatubo-full experiment. ECS stands for Equilibrium Climate Sensitivity.**


**Figure 1: Monthly mean aerosol optical depth at 550 nm due to stratospheric volcanic aerosols (variable aod550volso4) averaged over the tropics (30°S-30°N) used in the volc-pinatubo-full experiments.**


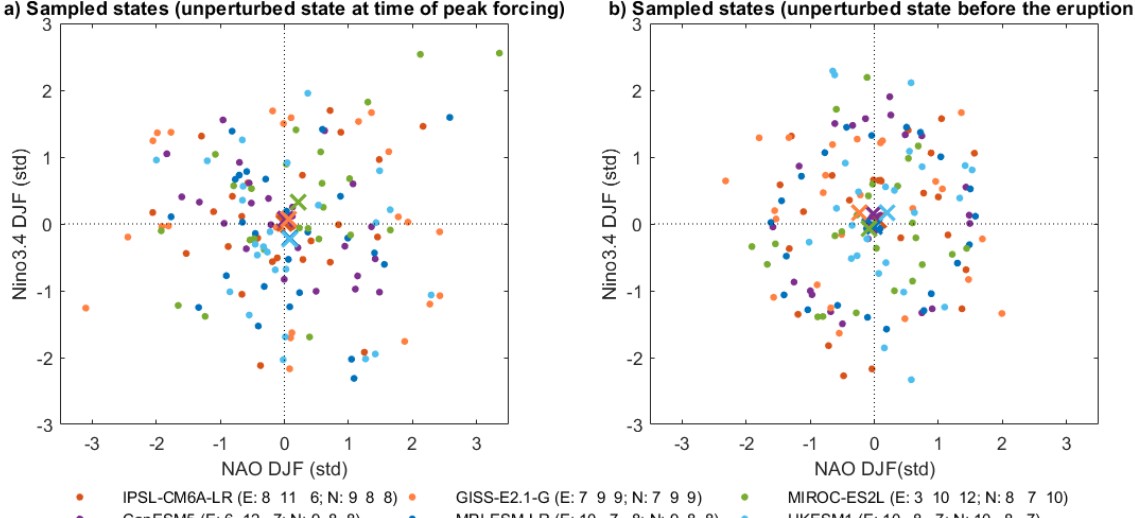

**Figure 2: Scatterplots of the sampled states of winter NAO and Niño3.4 indices across the participating models. According to the VolMIP protocol, the sampled states in the piControl correspond to the first post-eruption winter in the volc-pinatubo-full simulations (DJF 1992) when there is a peak in the prescribed forcing dataset (panel a). Panel b shows sampled states during the winter preceding the eruption (DJF 1991). Each coloured dot corresponds to one out of the 25 realizations for each model, crosses correspond to ensemble means. The legend at the bottom shows the colour corresponding to each model and the number of realizations pertaining to the three ranges (negative/cold; neutral; positive/warm) of each variability mode (E: ENSO; N: NAO). For both variability modes, the shown values are from indices obtained by standardization of the original December-January-February average time series over the whole piControl simulation.**

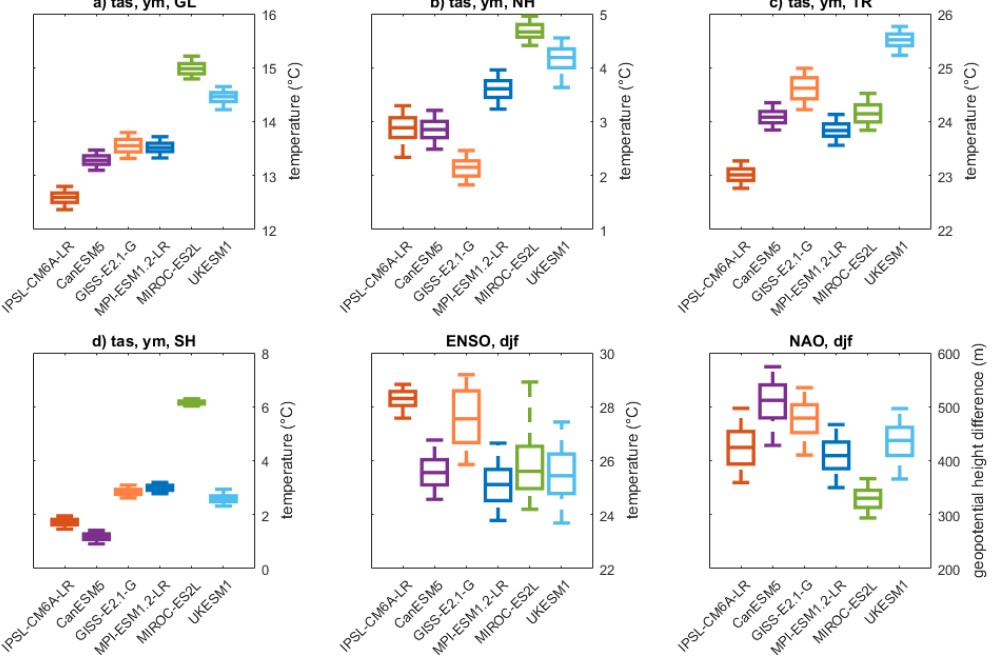

**Figure 3: Mean state and variability of unperturbed climates of the participating models. Distributions of selected climatic parameters are shown as Box-Whisker plots (median, 25th-75th and 5th-95th percentile ranges) for the piControl. Panels a-d refers to near-surface annual-mean (ym) near-surface air temperature (tas) spatially averaged over the whole globe (GL), the Northern Hemisphere extratropics (NH), the tropics (TR) and the Southern Hemisphere extratropics (SH). Following the VolMIP protocol, NAO is shown as the difference between the 500 hPa geopotential height in the two boxes that are used for the calculation of the index (see methods); ENSO is the average sea-surface temperature in the Niño3.4 region. Both indices are winter-average (December to February) time series.**



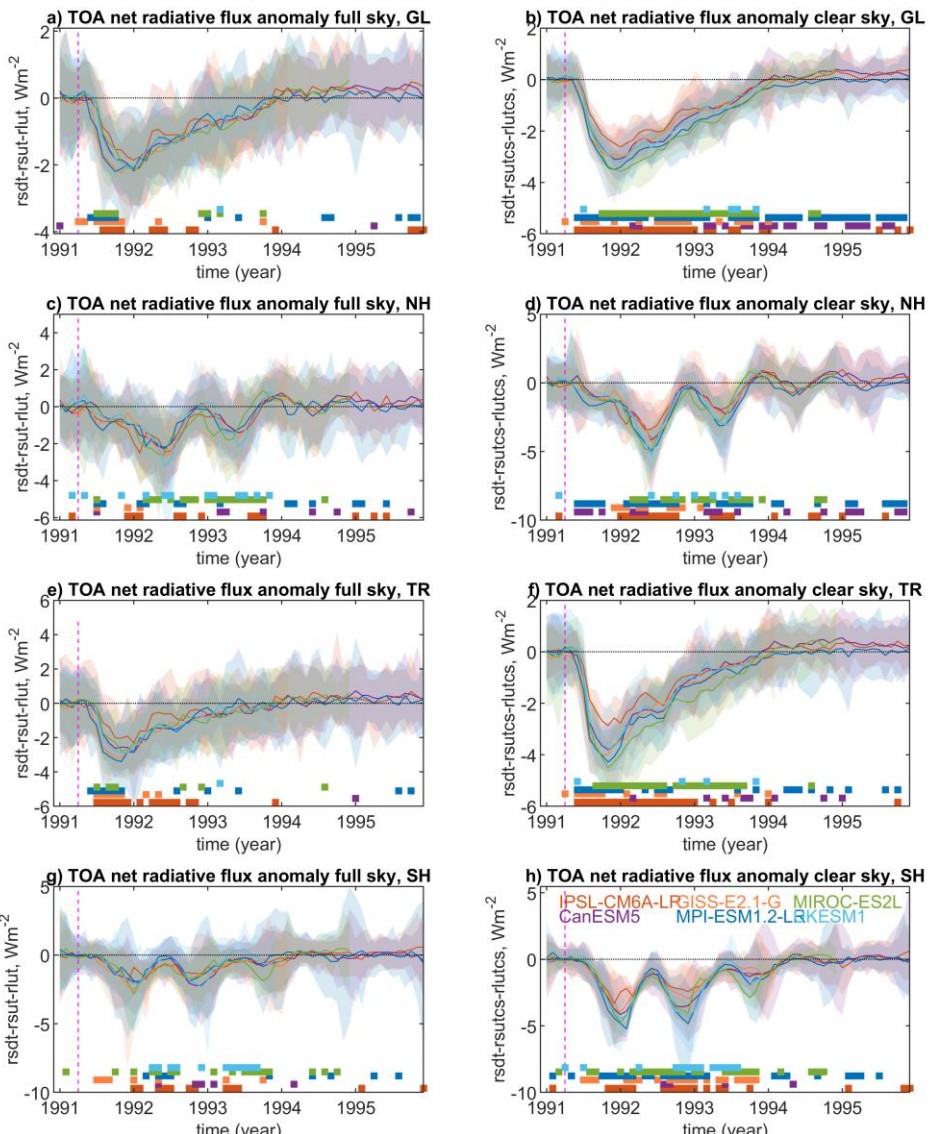

**Figure 4: Top-of-atmosphere (TOA) net vertical radiative flux anomalies under full sky (left panels) and clear sky (right panels) conditions in the volc-pinatubo-full multi-model ensemble for a,b) global (GL), c,d) Northern Hemisphere extratropical (NH) , e,f) tropical (TR) and g,h) Southern Hemisphere extratropical (SH) mean. Full-sky anomalies are calculated as incoming shortwave (rsdt) minus outgoing shortwave (rsut) minus outgoing longwave radiation (rlut); clear-sky anomalies are calculated as rsdt minus clear-sky outgoing shortwave (rsutcs) minus clear-sky outgoing longwave radiation (rlutcs). For each model the shading illustrates the ensemble envelope and the line the ensemble mean. Positive anomalies indicate increased downward flux. Squares at the bottom indicate when one model output is significantly different (p<0.05) from the ensemble members of all the other models according to the Mann-Whitney U test. The vertical dashed magenta line indicates the approximate timing of the eruption.**

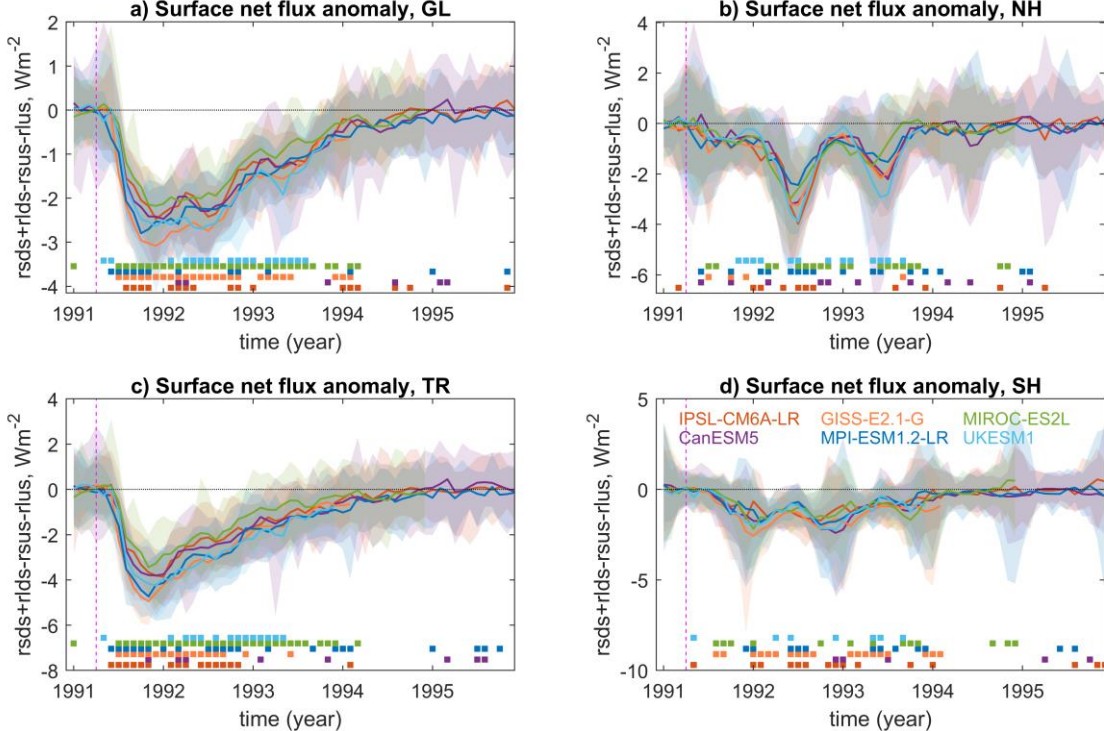

**Figure 5: Surface net vertical radiative flux anomalies in the volc-pinatubo-full multi-model ensemble for a) global (GL), b) Northern Hemisphere extratropical (NH) , c) tropical (TR) and d) Southern Hemisphere extratropical (SH) mean. The net flux anomalies are calculated as downward shortwave (rsds) plus downward longwave (rlds) minus upward shortwave (rsus) minus upward longwave radiation(rlus). For each model the shading illustrates the ensemble envelope and the line the ensemble mean. Positive anomalies indicate increased downward flux. Squares at the bottom indicate when one model output is significantly different (p<0.05) from the ensemble members of all the other models according to the Mann-Whitney U test. The vertical dashed magenta line indicates the approximate timing of the eruption.**

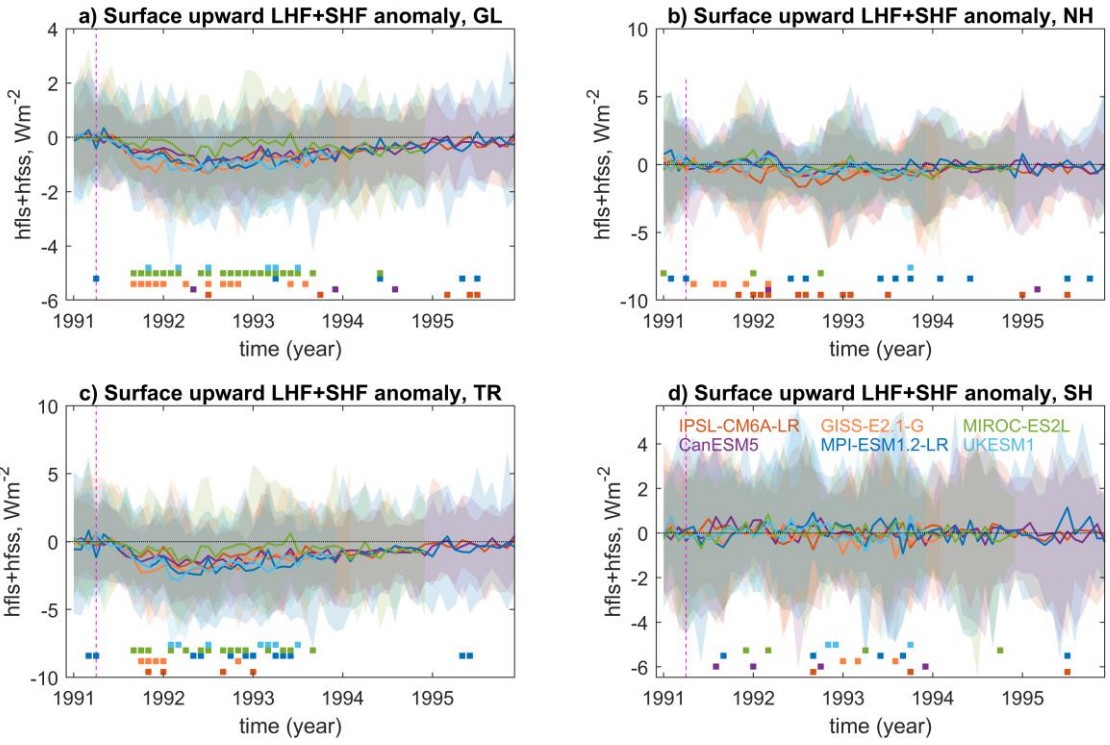


**Figure 6: As in Fig. 5 but for surface upward vertical latent heat flux (LHF) plus sensible heat flux (SHF) (or hfls+hfss).**

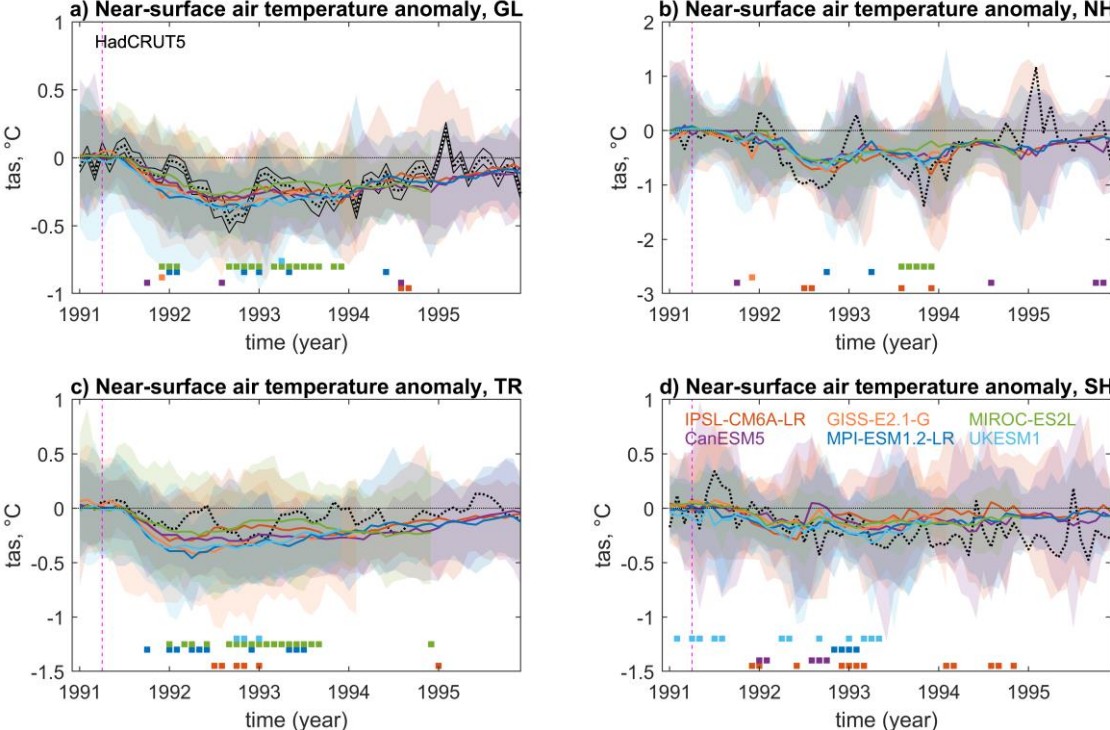

**Figure 7: As in Fig. 5 but for near-surface air temperature (tas). Observed HadCRUT anomalies (ensemble-means as deviations from the 1990 average calculated from detrended time series, see methods) are plotted as dotted black lines for comparison; uncertainty estimates are also provided for GL (continuous black line).**


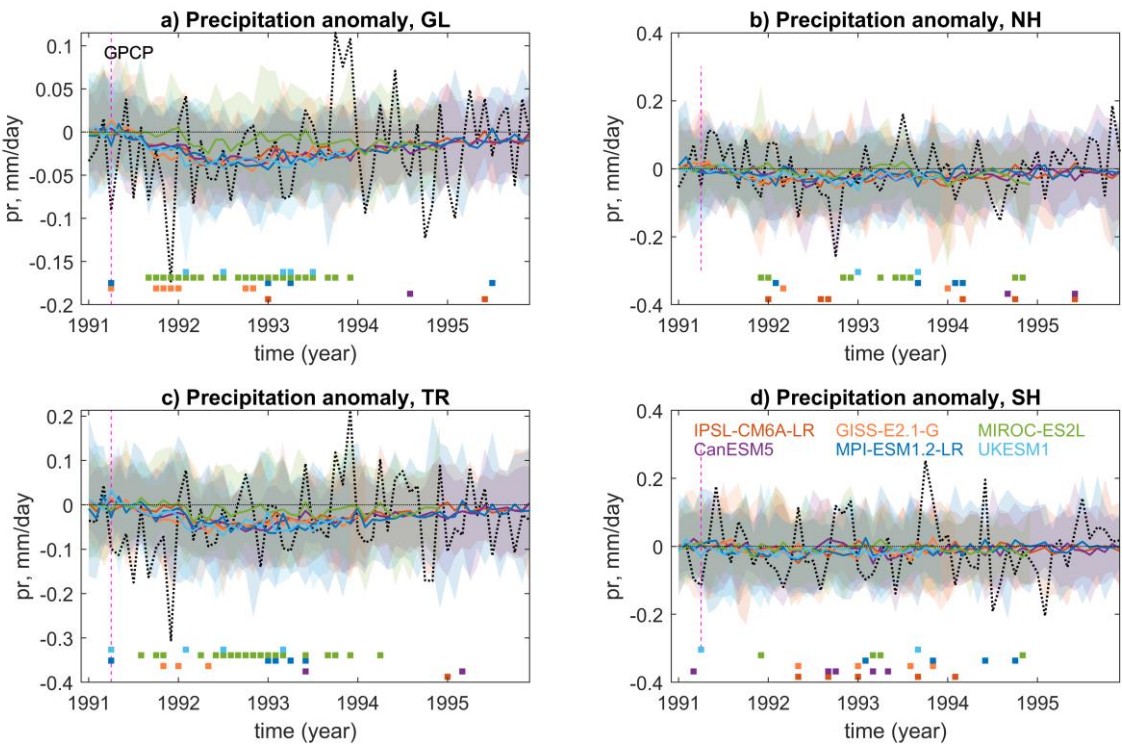

**Figure 8: As in Fig. 5 but for anomalies of total precipitation (pr). Observed GPCP anomalies (ensemble-means as deviations from the 1990 average calculated from detrended time series, see methods) are plotted as dotted black lines for comparison.**


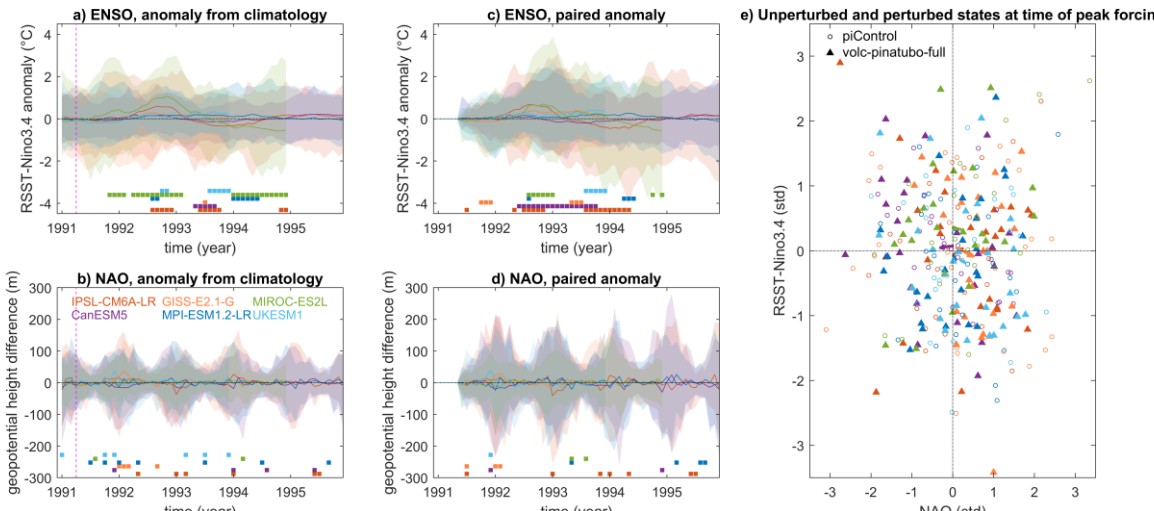

**Figure 9: ENSO (RSST-Nino3.4 index) and NAO anomalies in the volc-pinatubo-full multi-model ensemble. (a-d) Anomalies for the indices calculated according to the VolMIP protocol as deviations from the unperturbed climatology (a,b) and as paired anomalies (c,d). For each model the shading illustrates the ensemble envelope and the line indicates the ensemble mean. Squares at the bottom indicate when one model output is significantly different (p<0.05) from the others according to the Mann-Whitney U test. (e) scatterplots of standardized indices for the winter (DJF) season at the time of peak forcing in the volc-pinatubo-full. Shown are also corresponding values under unperturbed conditions following the VolMIP sampling protocol (see methods and Figure 2). Arrows indicate ensemble mean changes between unperturbed and volcanically perturbed anomalies.**

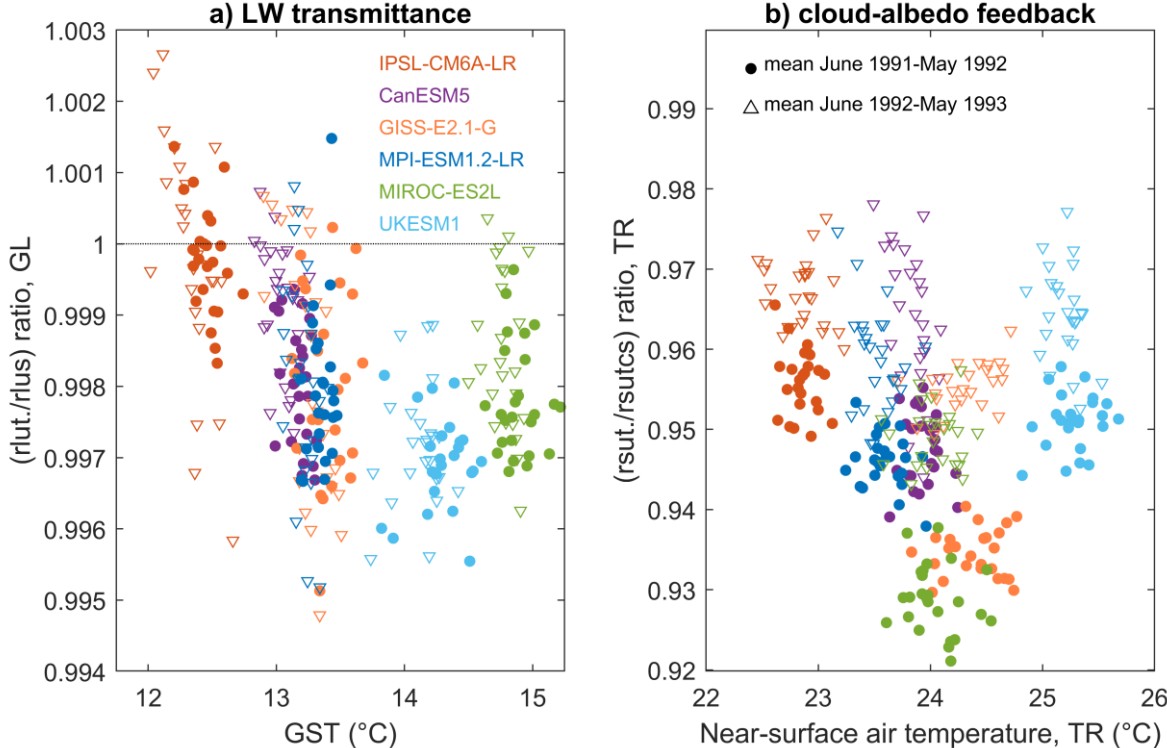

**Figure 10: Response of the atmospheric LW transmittance and of the cloud-albedo feedback. a) scatterplot of the ratio of the global-average LWt/LWs↑ (or rlut/rlus) calculated for the volc-pinatubo-full simulations over the corresponding piControl sections versus the post-eruption global-mean surface temperature (GST). b) scatterplot of the ratio of SWt/SWtcs for the tropical region (rsut/rsutcs) calculated for the volc-pinatubo-full simulations over the corresponding piControl sections versus the tropical-mean surface temperature. Analysis is based on averages for the periods from June 1991 to May 1992 (filled circles) and from June 1992 to May 1993 (empty triangles).**

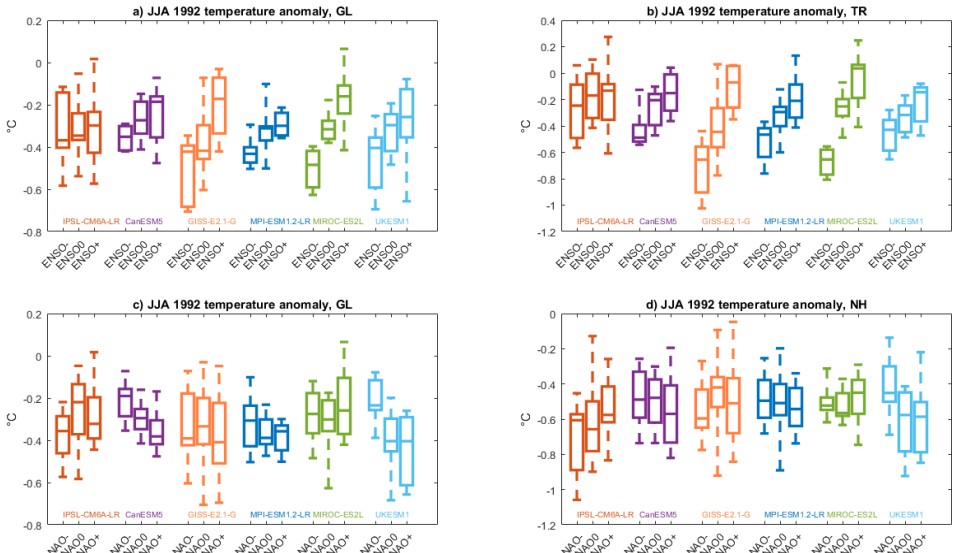

**Figure 11: Box-whisker plots of near-surface air temperature anomalies of selected areas for the second post-eruption boreal summer (JJA 1992) in the volc-pinatubo-full multi-model ensemble against the sampling conditions identified by the VolMIP protocol. Sampling conditions are identified by standardized winter-average (DJF) Nino3.4 and NAO states (positive if >0.5, negative if <-0.5, neutral/zero if in between) during the first post-eruption winter under unperturbed conditions (see Sect. 2.2). These states are accordingly used to cluster the temperature anomalies.**

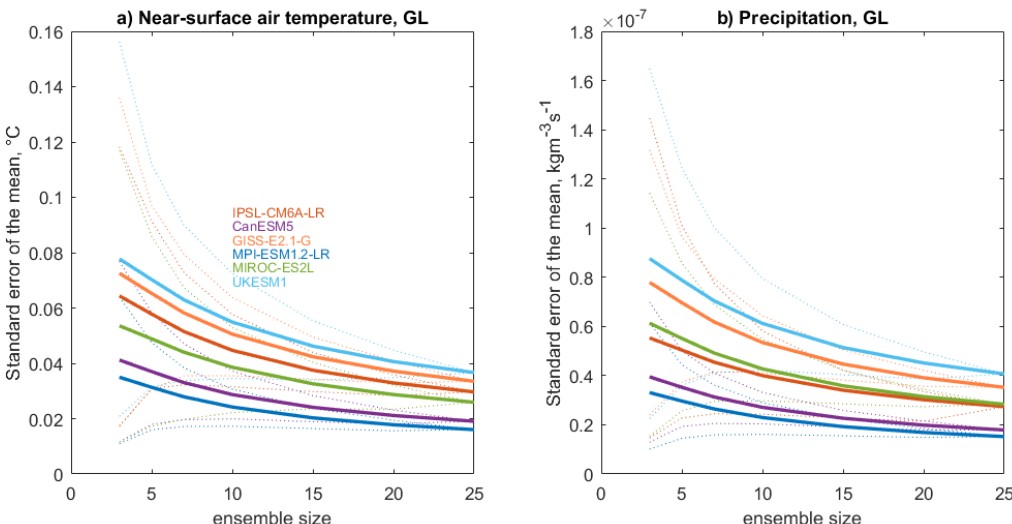

**Figure 12: Ensemble size and spread. Standard error of the mean of anomalies calculated for different ensemble sizes across the different models for two key surface variables: global-mean near-surface air temperature (a) and global-mean precipitation (b). Shown are means (thick line) and 5-95th percentiles ranges (see Sect. 2.2).**

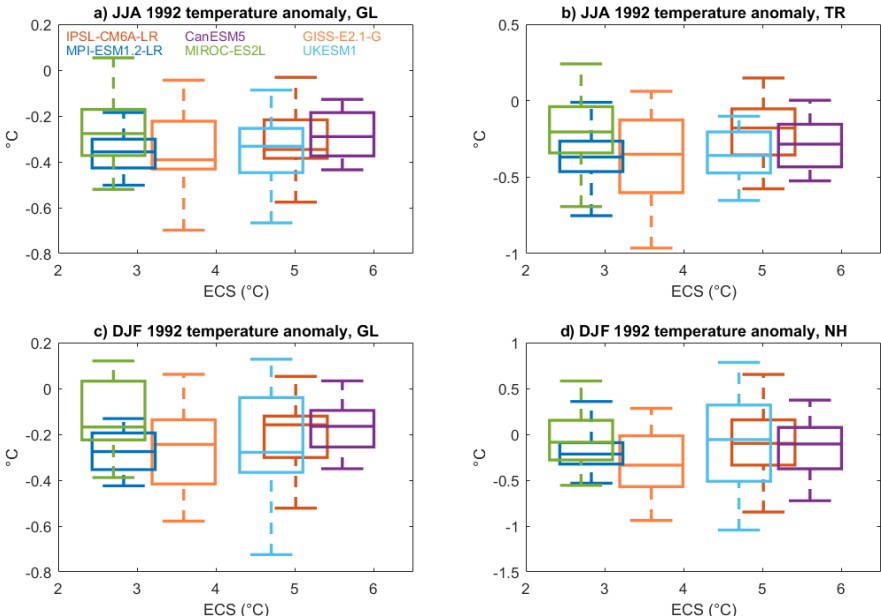

**Figure 13: Box-whisker plots of simulated post-eruption near-surface air temperature anomalies in the volc-piantubo-full simulations as function of equilibrium climate sensitivity (ECS). a,b) global-mean (GL) and tropical-mean (TR) anomalies for the second post-eruption boreal summer; c,d) GL and Northern Hemisphere-mean (NH) anomalies for the second post-eruption boreal winter.**