# Peer review of "Effects of forcing differences and initial conditions on inter-model agreement in the VolMIP volc-pinatubo-full experiment"

_Geoscientific Model Development, 2021_

## Author Response (AR1)

**Response to Anonymous Referee #1**

We thank Referee #1 for her/his helpful comments on our manuscript. We report in italics relevant comments by the referee.

*No observed values are used when evaluating model climatology and responses to volcanic forcing. Although I agree that the aim of this paper is to provide an initial assessment based on idealized experiments, not historical transient experiments which are comparable to the observations, assessing the degree of inter-model agreement in volcanic influences without any relevant comparison with observed values could be misleading given that models may have systematic biases. I strongly suggest including observed values somehow in their plots and interpreting results accordingly*

We agree with the referee that a comparison with observed climate anomalies around the Pinatubo eruption is a worthy addition to our analysis. In the revised manuscript we have therefore included observed anomalies in revised Figure 7 and Figure 8 and discuss how our multi-model results compare with observations (see section 3 of revised manuscript for a description of the employed data and section 4.3 of revised manuscript for the results). Overall, we found – in agreement with current knowledge - that the observed post-Pinatubo global cooling compares well with expected responses found in the volc-pinatubo-full ensemble, but internal variability causes increased deviations between observations and ensemble means when regional anomalies are considered. We also expanded on this by performing an analysis on a sub-ensemble where the pre-eruption state of ENSO is consistent with observations (see new Figures S12 and S13). This is discussed in detail in section 4 of the revised manuscript.

*This study aims at providing preliminary assessments but more efforts to quantify factors responsible for inter-model discrepancies would be useful. One way would be to add summary bar graphs or tables for some key variables (with observed estimates if possible, see my comment above) where readers can find actual values for individual models and how much differences exist between models and also between different ocean initial conditions (ENSO phases). Mostly, time series are displayed and it is inconvenient to identify specific model responses.*

In the revised manuscript we have included two supplementary tables. Supplementary Table S1 provides further details about the simulations compared to what was provided in the original manuscript, including relevant metadata such as the timing of branching of each realization from the piControl. The time series used in the manuscript are publicly available, so that quantitative estimates can be easily calculated in follow-up studies. Information in Table S1 will foster the use in follow-up studies of the dataset associated with this study. Supplementary Tables S2 and S3 provide estimates for global-mean near-surface temperature and precipitation responses for JJA 1992 for all realizations for all models, from

which it is possible to calculate diagnostics such as ensemble-mean and ensemble spread. Table S2 also highlights the sampled pre-conditions for the ENSO state.

*Some places need more explanations for better understanding. It's unclear how authors have selected samples for "equally distributed cold/neutral/warm states of ENSO and negative/neutral/positive states of NAO". Exact details of sampling methods look very important for interpreting results as well as for planning the next VolMIP protocol. Also, authors consider radiation feedbacks in their evaluations but its association with inter-model spreads needs to be explained more clearly. Another one is why ECS is considered here, which represents equilibrium sensitivity to doubled CO2.*

In the revised manuscript we have provided further metadata regarding the simulations in Supplementary Table S1, which also includes the pre-conditioning states sampled for the different participating models. We provide more information in the revised text about the VolMIP protocol regarding the sampling of initial conditions and have revised Figure 2 to include information about how the sampled ENSO and NAO state are distributed around the three ranges defined by the VolMIP protocol (i.e., first, second and third terciles of the index). The VolMIP protocol did not provide a specific algorithm for the sampling, so different groups proceeded differently, sometimes with a subjective approach. We have also revised the manuscript by recommending that an algorithm is developed and shared to ensure sampling of initial conditions is consistent across models.

In the revised manuscript we have introduced ECS in section 2.2 by moving the text there and rephrasing a sentence originally in section 5.2. The revised text reads as follows:

"Among the reported characteristics is each model's equilibrium climate sensitivity (ECS). ECS is relevant here due to a rich debate in the scientific literature about the use of climate anomalies after volcanic eruptions to infer equilibrium climate sensitivity (ECS) or transient climate sensitivity (e.g., Wigley et al., 2005; Boer et al., 2007; Merlis et al., 2014)"

*Authors conclude that influence of ocean initial conditions is weak or even negligible but this conclusion can be dependent on how to measure ENSO-like responses. Other studies used relative SST as authors briefly mentioned, and results can be affected much by applying different metrics. Since understanding ENSO influence is one of major issues, I think that adding more discussion with appropriate sensitivity tests would be useful, e.g. comparing relative SST responses with Nino3.4 responses. In terms of NAO or AO responses, target season and region can be revised as boreal winter and high latitude areas, for better comparisons with previous findings.*

We agree that relative SSTs are a better basis than absolute SSTs to calculate the Nino3.4 index and capture ENSO-like responses. In fact, we used the Nino3.4 index as defined by the VolMIP protocol. This is one of the aspects where there is room for improvement in a

possible second phase of the initiative. Therefore, we revised the original analysis based on absolute SSTs by using a Nino3.4 index calculated from relative SSTs (RSST-Nino3.4). There are obviously some differences in the results obtained from RSST-Nino3.4 and from Nino3.4, which particularly concern the early development and the clustering of models in two groups, one showing a warm ENSO response in the second post-eruption winter and a cold ENSO phase thereafter, and another showing a neutral (or lack of) ENSO response. Overall, our conclusions do not change regarding the importance of accounting for sampling biases (again seen especially for MIROC-ES2L) and the use of paired anomalies. Please see the substantially revised paragraph regarding ENSO in section 4.3 and the revised figures 9 and S8.

Concerning the NAO, we acknowledge that there is much room for improvement. However, we decided not to add additional analyses specific for the NAO in this study and devote further analyses to follow-up studies. We have added the following sentence in revised section 5.1: "Also, for the NAO, analyses focused on the winter season, when the mode is most stable, and in association with the more hemispheric-scale Arctic Oscillation can improve the detection of volcanic signals."

**Response to Anonymous Referee #2**

We thank Referee #2 for her/his helpful comments on our manuscript. We report in italics relevant comments by the referee.

Concerning the specific discussion points raised by the referee:

*I understand the benefits to build a multi-model protocol in which the volcanic forcing is commonly defined, to allow a "good agreement between the different models » and highlight the differences between the models in terms of response to external forcing independently to the implementation of the forcing. Overall, is there any risk that the homogenisation of the climate models encouraged in model inter-comparisons support the building of a unique family of similar models, all of them showing the same uncertainties that would be therefore more difficult to estimate? Should we encourage more contrasted model developments for a better understanding of the processes at play?*

This is a very useful comment. Multi-model intercomparisons allows for proper comparability across results from different models. By setting a strict experimental design it is more likely that any difference in model output can be understood. This may lead to homogeneization of climate models, but a more certain expected outcome is general improvement of models and sound interpretation of their results. The aim of VolMIP is not to achieve a model consensus or to homogenize models and model development, but to highlight uncertainties and understand the mechanisms that govern the climate response to volcanic forcing. The key point of using a consistent forcing across models in terms of aerosol radiative properties is explained in the VolMIP original paper (Zanchettin et al., 2016). In brief, this allows us to focus on how models differ in the climate response, as uncertainties generated by aerosol chemical and microphysical properties are neglected. In fact, in this sense VolMIP is a companion to the SPARC/SSiRC Interactive Stratospheric Aerosol Model Intercomparison project (ISA-MIP, Timmreck et al., 2018)) which covers the uncertainties in the pathway from the eruption source to the volcanic radiative forcing. Specifically, the aim of ISA-MIP is to constrain and to improve global aerosol models by using a range of observations to reduce the forcing uncertainties.  Both initiatives are supposed to interact to progress our understanding of how climate responds to strong volcanic eruptions. Of course, model agreement does not necessarily imply agreement with observations. Therefore, also to account for a comment by Referee #1, we have included a comparison with observed anomalies (see revised Figures 7 and 8 and new supplementary Figures S12 and S13). The results suggest that overall observations compare well with expected (i.e., ensemble mean) responses especially for global-mean surface temperature. We discuss implications for regional responses and for precipitation in the revised manuscript in section 4.3 and concerning the role of initial conditions in Section 5.2.

*Could we expect an impact of the anthropogenic forcings on the climate response to volcanic eruptions? In other words, would we expect different conclusions starting the volc-pinatubo-full experiment from control experiments produced with constant anthropogenic forcings corresponding to those observed at the beginning of the XXIth century, and/or in transient forcing experiments?*

Several studies point to the fact that background climate conditions can affect the climate response to volcanic forcing. We consider the volc-pinatubo experiments idealized exactly because the forcing is realistic, but the background climate state (from the unperturbed piControl simulation) is different from the actual climate state during the 1991 Pinatubo eruption. In the original manuscript, we even illustrate some details about the mean climate state and variability in piControl simulated by the different models as a possible source of inter-model differences. Therefore, we agree with the referee that conclusions might differ if the background climate state differs and/or in transient conditions, i.e., in presence of additional forcing agents. The potential of a warmer background climate state and presence of additional forcing to affect the climate response to a Pinatubo-like eruption is better stressed in the revised manuscript also by including a comparative analysis with observations for the 1991 Pinatubo eruption (revised Figures 7 and 8, and new supplementary Figures S12 and S13). The results show that at least for the global-mean surface temperature response observations match well with the expected responses, in agreement with Pauling et al. (2021). However, we consider these results only preliminary and stress the need for follow up studies to gain robust conclusions, especially as far as regional climates and the less understood precipitation response is concerned.

*P4, L.125-130: How do we deal the fact that the modes of variability typically show different patterns for the different models? Why not having considered an EOF approach, using for each model its mode of variability with its specific pattern?*

We dealt with it in the definition phase of the VolMIP protocol, when we choose to use box-based indices over EOF-based ones because our expectation is that total variability is separated differently into principal components in different models, which would add another level of uncertainty. We agree that there is an intrinsic problem in the use of predefined indices based on mathematical constructs (being EOF or box-based indices) rather than physical understanding. We have clarified this in the revised discussion by adding the following sentence: "More generally, given the variable representation of ENSO - and other climatic modes as well - in different climate models, the choice of the associated index should reflect physical understanding of the climate mode rather than merely build on a mathematical construct".

*P5, Characteristics of volc-pinatubo-full, the multi-model ensemble: Would it be possible to give more details about the spectral resolution of the models? Does it differ among the*

*models? The way that the VOLMIP forcing is distributed over the spectral bands could be detailed. More information about the vertical distribution of the forcing in the different models would be also welcomed: is the forcing vertically distributed in a stationary way, using monthly climatologies of the elevation of the atmospheric layers, or is the forcing vertically distributed on-line? I saw that this weakness is discussed at the end of the paper, but why not including more information about these model features in this publication?*

The spectral resolution does vary between the models. Unfortunately, there is no publication describing the CMIP6 volcanic forcing from the team at ETH that has produced the forcing, and so the exact details of the method that is used to distribute the forcing over the spectral bands is somewhat undefined. (It is clear the technique involves assuming a log-normal size distribution and using the multi-wavelength measurements from SAGE II to estimate size parameters, from which extinction and other optical properties can be calculated using Mie theory, but without a proper citation we hesitate to include this level of detail here.) We have added a sentence to the experiment description as follows: "The recommended CMIP6 stratospheric aerosol forcing is tailored to the spectral resolution of each participating model, which can vary greatly. For the models included in this study, the number of solar bands ranges from 4 to 23, and the number of terrestrial bands ranges from 9 to 16." Concerning the vertical distribution of the aerosol forcing, we prefer to leave the description and discussion of the technical details of vertical interpolation and truncating at the tropopause to future work.

*Initial conditions: Why not sampling the QBO on a similar way as the other modes? QBO impact on climate response to volcanic forcing has been evidence in Thomas et al., 2009. (Thomas, M. A., Giorgetta, M. A., Timmreck, C., Graf, H.-F., and Stenchikov, G.: Simulation of the climate impact of Mt. Pinatubo eruption using ECHAM5 – Part 2: Sensitivity to the phase of the QBO and ENSO, Atmos. Chem. Phys., 9, 3001–3009, https://doi.org/10.5194/acp-9-3001-2009, 2009). At lower frequency, why not considering different states of the AMV that might affect also the response of the modes of variability (Ménégoz, M., Cassou, C., Swingedouw, D., Ruprich-Robert, Y., Bretonnière, P.A. and Doblas-Reyes, F., 2018. Role of the Atlantic Multidecadal Variability in modulating the climate response to a Pinatubo-like volcanic eruption. Climate Dynamics, 51(5), pp.1863-1883). This point is discussed at the end of the article. Nevertheless, we do not know the reasons for which these modes have not been considered in the first edition of VOLMIP*

We clarify in the revised manuscript in section 2.1 that "The choice to limit the number of sampled climate modes to two is due to restrictions related to the ensemble size. ENSO and NAO were chosen due to their relevance in the scientific literature concerned with volcanically-forced climate variability when the VolMIP protocol was defined".

We agree that the state of QBO is a potential influencing factor on the climate response to volcanic eruptions. Since not all models spontaneously generate a QBO, we decided to not include it as a requirement for sampling in the final VolMIP protocol. The original VolMIP paper states that "volcanic radiative sampling of an eastern phase of the Quasi-Biennial Oscillation (QBO), as observed after the 1991 Pinatubo eruption, is preferred for those models that spontaneously generate such mode of stratospheric variability." The QBO was already discussed in the original manuscript in section 5.2. We have revised this part adding citation to the Hommel et al. (2015) and the suggested reference in the revised manuscript to support the statement that QBO affects volcanic forcing and the climate response to volcanic forcing. The part now reads as follows: "Among potentially relevant modes, the Quasi Biennial Oscillation (QBO) (Thomas et al., 2009) did not have explicit focus in VolMIP, but it is arguable that its representation in climate models will continue improving. However, the prescribed aerosol optical properties at the basis of VolMIP constitute a major limitation to an effective implementation of a sampling strategy for the QBO, since the phase of the QBO affects stratospheric transport including that of stratospheric aerosol, hence ultimately the volcanic forcing (e.g., Hommel et al., 2015). The effects of inconsistencies between QBO and prescribed forcing on volc-pinatubo experiments are unknown. Until this gap of knowledge is filled, we recommend continuing to sample an easterly phase of the QBO at the time of the eruption whenever possible". We also mention the QBO in the context of the connection between VolMIP and ISA-MIP at the end of revised paragraph 5.2.

Concerning the AMV, this is certainly of interest. The problem is how the increase in the number of variables used for the sampling affects the ensemble size. We need a balance, and for the short-term scales that are the focus of volc-pinatubo experiments we decided to opt for the NAO as a descriptor of the North Atlantic state. For the volc-long experiment of VolMIP that focus on the multiannual to decadal-scale climate response to volcanic forcing, we use the AMOC as reference index for sampling initial conditions. We have elaborated further on this in the revised manuscript, including the suggested reference Ménégoz et al. (2019) and a perspective on the implications of the coupling between AMV and AMOC suggested in the recent literature. We have revised the text as follows: "Decomposition of the AMV signal in paleoclimate simulations also suggests that the internal component of the AMV, which is tightly connected with the Atlantic meridional overturning circulation, lacks robust behaviour across simulations during periods of major volcanic forcing (Fang et al., 2021). VolMIP experiments are well suited to test such hypothesis. Therefore, the possibility to include AMV and Pacific Decadal Oscillation in the sampling protocol should be considered […]"

*P10: the ENSO differences among the models based on a temperature average over the Niño 3.4 area only might be affected by the ENSO specific position in each model. The ENSO*

*signature in models is often shifted Southward/Northward Eastward/Westward as compared to the observations, and it differs clearly from one model to another one. This could be discussed in the article. The same issue can be highlighted for the NAO signature, and this might be a much more important issue considering the typical spatial biases of the NAO pattern in the current generation of AOGCMs.*

We have elaborated further on the ENSO index, also following a comment by Referee #1. As highlighted in a point response above, inter-model differences and biases with respect to observations make it difficult to identify optimal indices that are expected to capture specific dynamics. We will discuss this better in the revised manuscript. At least for the NAO a recent paper using the same index definition employed in VolMIP suggests a marked consistency across CMIP6 models (Cusinato et al., 2021). Text in section 5.2 has been revised as follows: "More generally, given the variable representation of ENSO - and other climatic modes as well - in different climate models, the choice of the associated index should reflect physical understanding of the climate mode rather than merely build on a mathematical construct. For the NAO, at least, a recent multi-model study using the same index definition employed in volc-pinatubo-full suggests a marked consistency across CMIP6 models (Cusinato et al., 2021). In the future, the choice of an index should be supported as much as possible by preliminary assessments of climate model biases."

*P13, L. 402: "dynamical responses may be masked by broad tropical radiative cooling effects » -> So why not considering a relative ENSO index (Nino3.4 tas minus tropical tas) as done in several publications (e.g. Khodri, M., Izumo, T., Vialard, J., Janicot, S., Cassou, C., Lengaigne, M., Mignot, J., Gastineau, G., Guilyardi, E., Lebas, N. and Robock, A., 2017. Tropical explosive volcanic eruptions can trigger El Niño by cooling tropical Africa. Nature communications, 8(1), pp.1-13.). This is discussed in the end of the article, but why not including directly such a "RENSO index" in the article?*

The choice of using absolute SSTs for the ENSO index was based on the original VolMIP protocol. We keep the original Nino3.4 index for the illustration of the initial conditions, as this is the basis upon which the simulations were built. However, also following a comment by Referee #1, we have revised the original analysis of the responses based on absolute SSTs by using a Nino3.4 index calculated from relative SSTs (RSST-Nino3.4). We do not find robust major differences in the effects of sampling if the RSST-Nino3.4 index is used instead of the Nino3.4 (see Supplementary Figures S10 and S11).

However, differences arise when the ENSO response is diagnosed from results obtained from RSST-Nino3.4 and from Nino3.4, which particularly concern the early development and the clustering of models in two groups, one showing a warm ENSO response in the second post-eruption winter and a cold ENSO phase thereafter, and another showing a neutral (or

lack of) ENSO response. Overall, our conclusions do not change regarding the importance of accounting for sampling biases (again seen especially for MIROC-ES2L) and the use of paired anomalies. Please see the substantially revised paragraph regarding ENSO in section 4.3 and the revised figures 9 and S8. We strongly believe that ENSO deserves a follow-up specific study beyond the initial results considered here, that are mostly focused on assessing the effectiveness of the VolMIP protocol.

*P13-14: feedbacks: more explanations about the LW and SW ratios would be welcomed, to allow a better understanding of the sign of the feedbacks (negative versus positive) as well as the processes that are suggested in this Section. It is delicate in particular to understand whether the LW and SW changes are related to aerosol changes or to changes in the atmospheric temperature.*

We have improved the description about the LW and SW diagnostics in the revised manuscript, having in mind that this analysis is only preliminary for a study using also the volc-pinatubo-strat/surf experiments to fully disentangle the direct responses and the feedbacks involved in post-eruption climate evolution. In particular, we have added the following text in revised section 3: "For instance, for atmospheric LW transmittance, a value below one of the abovementioned ratio implies $(LW_t/LW_s\uparrow)$volc-pinatubo-full being less than $(LW_t/LW_s\uparrow)$piControl , hence a decrease of atmospheric LW transmittance under volcanic forcing compared to unperturbed conditions. These diagnostics integrates the effects of diverse processes, including direct radiative responses forced by the volcanic aerosol and feedbacks operating through changes in the global-mean surface temperature. Hence, they do not allow for a separation between direct and indirect effects of volcanic forcing, which requires additional experiments such as volc-pinatubo-surf/strat."

*Technical corrections:*

*L71: "sensitivity experiments aimed" -> which one? Maybe more information could be given here.*

We have included the names of the experiments and link to the corresponding descriptive section in the VolMIP paper.

*P6, L. 165: there is a more recent description of the Orchidee surface scheme in Cheruy et al., 2020 (Cheruy, F., Ducharne, A., Hourdin, F., Musat, I., Vignon, É., Gastineau, G., Bastrikov, V., Vuichard, N., Diallo, B., Dufresne, J.L. and Ghattas, J., 2020. Improved near‐surface continental climate in IPSL‐CM6A‐LR by combined evolutions of atmospheric and land surface physics. Journal of Advances in Modeling Earth Systems, 12(10), p.e2019MS002005.)*

Thanks, we have added the suggested paper in our references

*P10, L. 294: why referring to Table 1 here?*

Information about the length of the piControl simulations was included in older versions of Table 1. We now provide this information (as well as other relevant metadata) in new Supplementary Table 1. We have updated the reference in the revised text accordingly.

*P10, L.302: It is stated that IPSL-CM6 is warmer in the tropical Pacific, but this is only verified in the Nino 3.4 domain, since it is cooler on average over the whole tropical area if I understand well Figure 3?*

Right, we have rephrased as "IPSL-CM6A-LR yields substantially warmer and less variable winter sea-surface temperature in the equatorial Central Pacific (Nino3.4 region) compared to other models".

*Figures 4-5-6-7-8-9: A vertical line could indicate the exact timing of the eruption.*

We have updated the figures by indicating the timing of eruption.

*Figure 10 caption: it is not totally clear whether the y axis show simply LWt/LWs↑ for one experiment (volc-pinatubo-full) or an anomaly difference between this experiment and the control (as mentioned in the text at Line 409).*

Thanks for this comment, the description was unclear. We specify now in the caption that what is shown are "scatterplot of the ratio of the global-average LWt/LWs↑ (or rlut/rlus) calculated for the volc-pinatubo-full simulations over the corresponding piControl sections versus the post-eruption global-mean surface temperature (GST). b) scatterplot of the ratio of SWt/SWtcs for the tropical region (rsut/rsutcs) calculated for the volc-pinatubo-full simulations over the corresponding piControl sections versus the tropical-mean surface temperature." We have improved the description of the diagnostics also in section 3 (see also our response to a general comment above) and changed the labelling of the y axes in this figure.

*P14, L. 414: "a tendential lowering » -> lowering with the time after the eruption?*

The statement was indeed confusing, it meant a lowering of the rsut/rsus ration in volc-pinatubo-full compared to piControl simulations. We have rephrased this to make it clear. Please see also the clarifications in other parts of this section.

*P15: 4.5 effect of sampling strategy: Again, it could be relevant to consider relative ENSO index (Nino 3.4 versus all the tropical areas) to disentangle the dynamical response of ENSO from the radiative cooling. The fact that the winter NAO does not affect the climate response to the volcanic forcing might be also explained by the relative small persistence of this mode of variability as compared to ENSO or AMV for example.*

We have elaborated further on the choice of the ENSO index, as also discussed in a response to a comment above. In summary, there is no robust major change in the sampling if RSST-Nino3.4 or Nino3.4 indices are used. The use of RSST compared to absolute SST instead matters for the ENSO response. We therefore decided to keep the original figure for ENSO in the revised manuscript as this stems for the sampling protocol used for VolMIP. Following the recommendation by the referee, we have however added Supplementary Figure S10 and S11 and discuss the choice of ENSO index in the revised text of section 4.5.

*P15: Ensemble size: In Figure 12, what is the period considered to compute the GST? (First year post-eruption?)*

GST response is calculated for the average of the year 1992. We have specified this in the revised manuscript.

*P16, L.480: compare -> compared*

Corrected

*P18, L. 560: to understanding -> to understand.*

Corrected

**Response to Chief Editor comment**

We thank the Chief Editor for the comment on our manuscript, which stimulated us to provide additional details about the models used in our study and clarify the availability of data during and after the revision process.

All six models used in our study are in their exact CMIP6 version as employed for the CMIP6-endorsed initiative VolMIP. We specify this in the revised manuscript and provide additional details about the models as well as the simulations (see in particular revised Sections 2, revised Table 1 and new Supplementary Table 1). We have revised the Code and Data Availability section accordingly and also included there a code availability statement for each model in the exact version employed.

Concerning the data availability, the raw data used in this study are part of the output of CMIP6, hence they are available through the ESGF for both the piControl and the volc-pinatubo-full experiments. We have included such statement in the revised Code and Data Availability section.

As stated in our manuscript, to facilitate the revision process and the public discussion of our manuscript we have provided the spatially-average data we have calculated from the gridded monthly output of each model in a long-term archive at: https://vesg.ipsl.upmc.fr/thredds/catalog/VOLMIP/volc-pinatubo-full/catalog.html

**References**

Cusinato, E., A. Rubino, and D. Zanchettin. Winter Euro-Atlantic Climate Modes: Future Scenarios From a CMIP6 Multi-Model Ensemble. Geophys. Res. Lett., 48, e2021GL094532, doi: https://doi.org/10.1029/2021GL094532, 2021

Fang, S.-W., Khodri, M., Timmreck, C., Zanchettin, D., and Jungclaus, J.H.: Disentangling Internal and External Contributions to Atlantic Multidecadal Variability Over the Past Millennium. Geophys. Res. Lett., 48 (23), e2021GL095990, https://doi.org/10.1029/2021GL095990, 2021.

Hommel, R., Timmreck, C., Giorgetta, M. A., and Graf, H. F.: Quasi-biennial oscillation of the tropical stratospheric aerosol layer, Atmos. Chem. Phys., 15, 5557–5584, https://doi.org/10.5194/acp-15-5557-2015, 2015.

Pauling, A. G., Bushuk, M., and Bitz, C. M.: Robust Inter-Hemispheric Asymmetry in the Response to Symmetric Volcanic Forcing in Model Large Ensembles, Geoph. Res. Lett., 48, e2021GL092558, https://doi.org/10.1029/2021GL092558, 2021

Thomas, M. A., Giorgetta, M. A., Timmreck, C., Graf, H.-F., and Stenchikov, G.: Simulation of the climate impact of Mt. Pinatubo eruption using ECHAM5 − Part 2: Sensitivity to the phase of the QBO and ENSO, Atmos. Chem. Phys., 9, 3001–3009, https://doi.org/10.5194/acp-9-3001-2009, 2009.

Timmreck, C., Mann, G. W., Aquila, V., Hommel, R., Lee, L. A., Schmidt, A., Brühl, C., Carn, S., Chin, M., Dhomse, S. S., Diehl, T., English, J. M., Mills, M. J., Neely, R., Sheng, J., Toohey, M., and Weisenstein, D.: The Interactive Stratospheric Aerosol Model Intercomparison Project (ISA-MIP): motivation and experimental design, Geosci. Model Dev., 11, 2581–2608, https://doi.org/10.5194/gmd-11-2581-2018, 2018.

Zanchettin, D., Khodri, M., Timmreck, C., Toohey, M., Schmidt, A., Gerber, E. P., Hegerl, G., Robock, A., Pausata, F. S. R., Ball, W. T., Bauer, S. E., Bekki, S., Dhomse, S. S., LeGrande, A. N., Mann, G. W., Marshall, L., Mills, M., Marchand, M., Niemeier, U., Poulain, V., Rozanov, E., Rubino, A., Stenke, A., Tsigaridis, K., and Tummon, F.: The Model Intercomparison Project on the climatic response to Volcanic forcing (VolMIP): experimental design and forcing input data for CMIP6, Geosci. Model Dev., 9, 2701–2719, https://doi.org/10.5194/gmd-9-2701-2016, 2016.